# Voltage control of magnetism in $Fe_{3-x}GeTe_2$/$In_2Se_3$ van der Waals ferromagnetic/ferroelectric heterostructures

Jaeun Eom [1,2], In Hak Lee[1], Jung Yun Kee[1,3], Minhyun Cho[4], Jeongdae Seo[5], Hoyoung Suh[6], Hyung-Jin Choi[7], Yumin Sim[8], Shuzhang Chen[9,10], Hye Jung Chang [6], Seung-Hyub Baek[7], Cedomir Petrovic [9,10], Hyejin Ryu [1], Chaun Jang[1], Young Duck Kim [4], Chan-Ho Yang [5], Maeng-Je Seong [8], Jin Hong Lee [1]✉, Se Young Park [3,11]✉ & Jun Woo Choi[1]✉

We investigate the voltage control of magnetism in a van der Waals (vdW) heterostructure device consisting of two distinct vdW materials, the ferromagnetic $Fe_{3-x}GeTe_2$ and the ferroelectric $In_2Se_3$. It is observed that gate voltages applied to the $Fe_{3-x}GeTe_2$/$In_2Se_3$ heterostructure device modulate the magnetic properties of $Fe_{3-x}GeTe_2$ with significant decrease in coercive field for both positive and negative voltages. Raman spectroscopy on the heterostructure device shows voltage-dependent increase in the in-plane $In_2Se_3$ and $Fe_{3-x}GeTe_2$ lattice constants for both voltage polarities. Thus, the voltage-dependent decrease in the $Fe_{3-x}GeTe_2$ coercive field, regardless of the gate voltage polarity, can be attributed to the presence of in-plane tensile strain. This is supported by density functional theory calculations showing tensile-strain-induced reduction of the magnetocrystalline anisotropy, which in turn decreases the coercive field. Our results demonstrate an effective method to realize low-power voltage-controlled vdW spintronic devices utilizing the magnetoelectric effect in vdW ferromagnetic/ferroelectric heterostructures.

The magnetoelectric effect is a classical topic in material science and physics. In magnetoelectric materials, the electric and magnetic properties are coupled: the magnetic moments (electric dipoles) can be manipulated by the electric (magnetic) field[1–12]. The interactions between the distinct magnetic and electric order parameters, along with the associated atomic, ionic, or molecular structure, can result in emergence of rich and intriguing physical phenomena. An effective strategy to achieve the magnetoelectric effect is to combine materials with distinct ferroic orders. For instance, in a ferromagnetic/ferroelectric composite system, the electric-field-controlled polarization or strain in the ferroelectric can induce modulation of the magnetic properties in the adjacent ferromagnet[5–12]. When utilized in nano-electronic devices, these composite systems offer multifunctional and advanced device characteristics that cannot be attained by the constituent materials independently[8,9,13].

[1]Center for Spintronics, Korea Institute of Science and Technology (KIST), Seoul 02792, Korea. [2]Department of Physics and Astronomy, Seoul National University, Seoul 08826, Korea. [3]Department of Physics, Soongsil University, Seoul 06978, Korea. [4]Department of Physics and Department of Information Display, Kyung Hee University, Seoul 02447, Korea. [5]Department of Physics, KAIST, Daejeon 34141, Korea. [6]Advanced Analysis Center, Korea Institute of Science and Technology (KIST), Seoul 02792, Republic of Korea. [7]Electronic Materials Research Center, Korea Institute of Science and Technology (KIST), Seoul 02792, Korea. [8]Department of Physics, Chung-Ang University, Seoul 06974, Korea. [9]Condensed Matter Physics and Materials Science Department, Brookhaven National Laboratory, Upton, NY 11973, USA. [10]Department of Physics and Astronomy, Stony Brook University, Stony Brook, NY 11794-3800, USA. [11]Origin of Matter and Evolution of Galaxies (OMEG) Institute, Soongsil University, Seoul 06978, Korea. ✉e-mail: jinhong.lee87@gmail.com; sp2829@ssu.ac.kr; junwoo@kist.re.kr

In this article, we experimentally demonstrate the magneto-electric effect in a van der Waals (vdW) material heterostructure, consisting of a ferromagnetic and a ferroelectric vdW material. Ferromagnetic/ferroelectric vdW heterostructures offer unique advantages over conventional thin-film-based composite systems to investigate the magnetoelectric effect. (1) Heterostructures composed of various families of vdW materials can be readily fabricated by mechanical exfoliation, transfer, and stacking, which makes it possible to investigate new functionalities arising from the interplay between symmetry-breaking orderings, that are separately present in the constituent materials, through interfacial coupling[14,15]. (2) Various fundamental material properties of two-dimensional (2D) vdW materials can be functionally controlled through strain, gating, proximity effect, and twist angle, which facilitates the controllability of vdW materials via external methods[16]. (3) The weak interlayer magnetic interactions make vdW ferromagnets an ideal material platform to investigate interfacial interactions, i.e., the coupling at the ferromagnetic/ferroelectric interface can play a more prominent role since the ferroic orderings within the component materials are not too strong. A recent study shows that interfacial magnetic effects can indeed be enhanced when the interlayer magnetic coupling is weak[17].

Given these advantages, it is natural to conceive a vdW-material-based device consisting of a ferromagnetic and a ferroelectric vdW material in which the change in the polarization by applied electric field can modify the properties of the ferromagnet by interfacial coupling. In fact, theoretical studies suggest the possibility of magneto-electric interactions between vdW materials[18–22]. Yet, the voltage-controlled magnetoelectric effect in vdW heterostructures, consisting of two functional vdW materials, has not been experimentally demonstrated so far.

Here, we study the voltage control of magnetism in a vdW heterostructure device consisting of a ferromagnetic $Fe_{3-x}GeTe_2$ ($x \approx 0.36$) layer and a ferroelectric $\alpha$-$In_2Se_3$ layer. $Fe_{3-x}GeTe_2$ (FGT), which is a hole-doped version of $Fe_3GeTe_2$[23,24], is a vdW ferromagnetic metal with bulk Curie temperature $T_C$ around 150 K and perpendicular magnetic anisotropy[25–27]. Large doping-dependent modulation of $T_C$ and coercive field is reported; these doping effects are closely related with the change in magnetocrystalline anisotropy induced by shifts in the energy bands[28]. This intricate relationship between electronic structures and magnetic anisotropy facilitates the capability to electrically control magnetism by changing the electronic structures, as the magnetic anisotropy acts as an energy barrier in the magnetization switching process. The non-centrosymmetric $\alpha$-$In_2Se_3$ (IS) is a vdW semiconductor that exhibits room temperature ferroelectricity, showing a relatively small ferroelectric coercivity of $\sim 2 \times 10^7 \, V \, m^{-1}$[29–32]. Due to the voltage-dependent expansion of in-plane lattice constants[33–35], a large piezoelectric effect is expected which allows the control of in-plane strain and surface charges by applied voltage.

In our experiments, the magnetic properties of FGT are measured while applying an external voltage to the FGT/IS vdW heterostructure, using a magneto-optic Kerr effect (MOKE) microscopy system. We find that the magnetic properties of FGT show a sizeable voltage-dependent change where the decrease in coercive field is observed in both positive and negative voltages. This unusual homopolar behavior is further investigated by voltage-dependent Raman microspectroscopy which confirms the existence of piezoelectric strain in the FGT/IS heterostructure providing tensile strain for both positive and negative voltages. The link between the strain and magnetic properties of FGT is investigated by first-principles density functional theory (DFT) showing substantial decrease in the magnetocrystalline anisotropy with tensile strain, which can account for the reduction of the magnetic coercivity. Our experiments and calculations demonstrate a new method of voltage-controlled magnetism via strain coupling in a vdW ferromagnetic/ferroelectric heterostructure. Furthermore, our realization of a vdW magnetoelectric device signals a

major advance towards developing low-power all-2D-material-based spintronic devices[36], e.g., 2D versions of energy-efficient magneto-electric spin–orbit (MESO) devices, recently proposed in the semiconductor industry[13].

## Results and discussion

### Device characteristics and magnetic properties

Figure 1a shows the schematic diagram of our vdW heterostructure. A ferroelectric IS flake is sandwiched between two metallic materials, FGT and graphite, where the ferromagnetic FGT also functions as a gate electrode. An optical microscopy image of the FGT/IS heterostructure is displayed in Fig. 1b (also see "Methods" section for details of the device fabrication process and layer thicknesses). In the scanning transmission electron microscopy (STEM) images of FGT/IS (Supplementary Information (SI) Fig. S1), we observe a thin (~2 nm) polymer buffer layer, unintentionally inserted between FGT and IS during the mechanical exfoliation process. This layer may in fact act as a flexible adhesive layer; earlier studies show that polymer layers can effectively transfer in-plane strain to an adjacent vdW materials[37–39], such that it may in fact help transfer in-plane strain from the IS to FGT in this work (see "Discussion" section).

A gate voltage ($V_G$) is applied across the IS between the FGT (top electrode) and graphite (bottom electrode); the top FGT is grounded such that negative (positive) $V_G$ denotes the FGT being at higher (lower) electric potential. The $V_G$-dependent gate current ($I_G$) flowing through the IS, measured under the illumination of laser at 70 K, exhibits a diode-like behavior (Fig. 1c). This originates from the Schottky barriers at the IS interfaces[30]. In addition, the asymmetry between the top- and bottom-electrode materials may enhance the rectification feature in our device[40]. The FGT magnetic properties, obtained by MOKE, show robust ferromagnetism below its $T_C$ (Fig. 1d). The square shaped out-of-plane magnetic hysteresis loop at zero bias

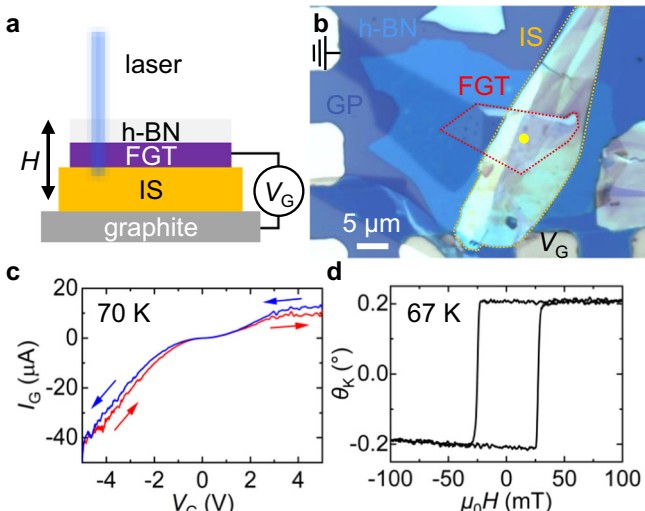

**Fig. 1 | Device characteristics and magnetic properties of the vdW heterostructure. a** Schematic illustration of the FGT/IS heterostructure showing the MOKE and Raman laser beam paths. The external magnetic field ($H$) is applied out-of-plane, and the gate voltage ($V_G$) is applied across the IS layer between the FGT and graphite. **b** Optical microscopy image of the heterostructure device. The FGT and IS flakes are indicated by dotted boundaries. The FGT is connected to an Au electrode located at the top-left corner via a graphite (GP) flake. See Methods for details of the device fabrication process and structure. The scale bar indicates 5 µm. **c** $I_G$–$V_G$ curves obtained at 70 K under the illumination of laser with the Raman measurement set-up. **d** A representative magnetic hysteresis loop of FGT at 67 K at zero bias ($V_G = 0$ V). The Kerr rotation angle ($\theta_K$) is linearly proportional to the FGT magnetization[64].

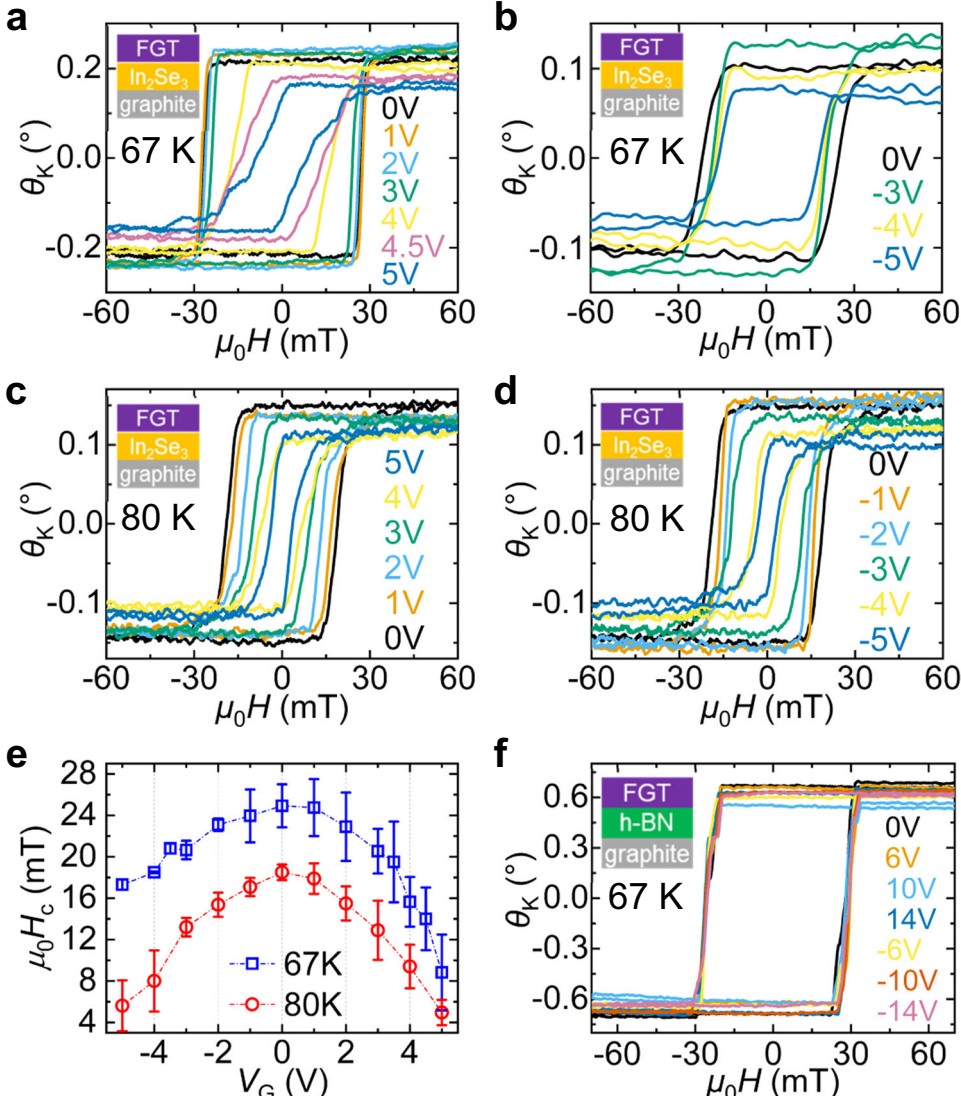

**Fig. 2 | Voltage-dependent magnetic properties measured by MOKE. a–d** The magnetic hysteresis loops of the FGT/IS heterostructure as a function of voltage, for (**a**) positive $V_G$ at 67 K, (**b**) negative $V_G$ at 67 K, (**c**) positive $V_G$ at 80 K, and (**d**) negative $V_G$ at 80 K. **e** The coercive field plotted as a function of $V_G$. The error bars represent the standard deviation of multiple measurement data. **f** The magnetic hysteresis loops of FGT/h-BN as a function of $V_G$. The plot legends are in sequential order, i.e., the measurements are performed as the magnitude of $V_G$ is progressively increased or decreased.

($V_G = 0$ V) is consistent with the perpendicular magnetic anisotropy of FGT reported previously[24,28,41,42].

## Voltage-dependent modulation of magnetic properties in the vdW heterostructure

Voltage-induced effects on the magnetic properties of the FGT/IS heterostructure are investigated by measuring voltage-dependent magnetic hysteresis loops. The MOKE signals were probed at the yellow spot in Fig. 1b, where the FGT overlaps the IS. The measurements were performed at $T = 67$ K and 80 K, below the ferromagnetic $T_C$ of FGT. The magnetic hysteresis of FGT show square-shaped loops at $V_G = 0$ V. Note the small coercive field (e.g., ≈30 mT at 67 K) of the FGT ($Fe_{2.64}GeTe_2$) used in this work, which is a hole-doped version of $Fe_3GeTe_2$. Upon voltage application, the FGT magnetic properties exhibit an intriguing $V_G$-dependent change (Fig. 2a–d). At positive $V_G$, the coercive field ($H_c$) decreases as $V_G$ increases (Fig. 2a, 2c) with this $V_G$-induced $H_c$ decrease becoming particularly prominent for large voltages ($V_G = 4$–5 V). For the negative $V_G$ case, $H_c$ again decreases as the magnitude of $V_G$ increases (Fig. 2b, 2d). In Fig. 2e, $H_c$ is plotted as a function of $V_G$, which illustrates the $V_G$-induced $H_c$ decrease. It is clear

that both positive and negative biases act to decrease $H_c$. This $V_G$-dependent $H_c$ decrease is consistently observed in other FGT/IS devices (SI Fig. S2). We also find that the voltage effect does not show FGT-thickness-dependence up to 20 nm (See SI Fig. S3 and discussion). To examine whether this voltage effect is reversible or remanent, we measure the hysteresis loops when $V_G$ is decreased back to zero after voltage application. Square-like loops with larger $H_c$ reappear at $V_G = 0$ (SI Fig. S4), implying that the $V_G$-induced $H_c$ modulation is reversible and mostly non-remanent. The reversible behavior is observed over 50 cycles of voltage application (SI Fig. S5), showing that this voltage-tunable magnetic effect is quite robust and stable.

In an effort to verify the role of the IS in the $V_G$-induced $H_c$ change in the vdW heterostructure, additional $V_G$-dependent MOKE measurements are performed with a FGT(12 nm)/h-BN(10 nm) heterostructure device at 67 K, in which a paraelectric and insulating h-BN replaces IS. Here, the loops are unaffected by $V_G$ up to ±14 V (Fig. 2f), showing that the voltage effect does not appear when h-BN is used for voltage gating. In another control experiment, MOKE measurements on a FGT/$\beta$-$In_2Se_3$ heterostructure device, in which a non-ferroelectric $\beta$-$In_2Se_3$ is used, again does not show significant change when $V_G$ is applied (SI Fig.

S6). These control experiments confirm that the presence of ferro-electric/piezoelectric IS ($\alpha$-In$_2$Se$_3$) adjacent to FGT is essential for the emergence of the voltage effect on the FGT magnetic properties.

Given the homopolar voltage dependence of magnetic properties and absence of voltage-dependent change in the centrosymmetric substrate, we propose the interfacial coupling that transfers the strain developed in IS to FGT as the origin of the observed non-remanent voltage effect in the magnetic properties of FGT. The Raman analysis and DFT calculations that follow will confirm this. Other possible interpretations, such as current-induced effects or interfacial charge from the out-of-plane polarization, are excluded based on our experimental results.

First, current-induced Joule heating effects can be ruled out from the lack of any voltage-dependent change in the $T_C$ of the FGT in the heterostructure devices. In SI Fig. S7, we observe that the $T_C$ of the FGT flakes are $\approx 120$ K at zero bias, and the $T_C$ shows little change with voltage application ($V_G = \pm 4$–5 V). In the discussions following the Raman analysis, we further rule out the possibility of current-induced Joule heating effects. Second, current-induced spin-torque effects can be excluded: the current density flowing through the FGT/IS hetero-junction is $\sim 10^6$ A m$^{-2}$, which is $\sim 5$ orders of magnitude smaller than the $\sim 10^{11}$ A m$^{-2}$ current density typically required for any significant spin-transfer-torque or spin–orbit-torque effects to occur. Additionally, we do not observe any electrically induced $H_c$ change in the region of the same FGT/IS heterostructure device where the ferroelectric IS is absent, even when the same level of current is flowing through the region (see SI Fig. S8). This effectively eliminates any possible current-induced spin-torque effects. Third, the effect of surface charge induced by the out-of-plane polarization of IS can be excluded since $H_c$ decreases for both positive and negative voltages (Fig. 2a–c). The effect of IS surface charge would be opposite when the voltage polarity is switched. Given the previous report of remanent effect from the polarization-induced IS surface charge[33], the non-remanent $V_G$-induced $H_c$ change (SI Fig. S4) provides further evidence that the IS surface charge is not likely responsible for the voltage effect. We also note that while the interfacial polymer layer might act as a dielectric resulting in accumulated interfacial charges induced by the IS polarization, the relatively thick metallic FGT thickness ($\sim 10$ nm), compared to the electron screening length, would prevent large electron/hole doping effects despite the possible presence of accumulated interface charges; earlier studies show that extremely large electron/hole doping, rarely achievable in solid state devices, is required for significant changes in the FGT magnetic anisotropy[28]. Deliberately adding a thin metallic charge dissipation layer underneath the FGT might help further rule out the effect of polarization-induced surface charge modulation; this is not trivial as such control experiment is based on assumptions that the metallic layer is stretchable enough for strain-transfer across all the interfaces.

The homopolar behavior of $H_c$ strongly suggests that the strain from the IS is the main driving force. The electric-field-induced strain modulation (or converse piezoelectric effect) of IS is identical regardless of voltage polarity as long as the external electric field ($\pm 6$ V in this work) is much larger than the ferroelectric switching field of IS (i.e., below $\sim 1$ V for the 50-nm-thick IS used in this work)[30,40]; in such condition, the polarization is always kept parallel to the external field such that the sign of the converse piezoelectric coefficient remains the same. The amplitude of in-plane lattice expansion/contraction is determined not by the direction but by the amplitude of the out-of-plane polarization. This change in the in-plane lattice constants provides strain to the FGT by interfacial coupling, which can induce the $V_G$-dependent $H_c$ modulation in the FGT/IS heterostructure.

## Voltage-induced tensile strain in the vdW heterostructure

In order to observe the voltage-dependent strain modulation in the heterostructure, its Raman spectra are measured as a function of $V_G$.

Raman micro-spectroscopy is a useful technique to study the local strain state of micrometer-sized flakes, since the characteristic wave-number of phonon modes are sensitive to lattice expansion/contraction[43–45]. Figure 3a shows the $V_G$-dependent Raman shift measured at 70 K, i.e., approximately the temperature at which the MOKE measurements are performed. We observe four peaks within the Raman shift range of 80–160 cm$^{-1}$ where the Raman modes at zero bias ($V_G = 0$ V) are indicated by vertical lines. This region of spectra allows easy tracking of both FGT and IS peaks depending on a voltage bias: two peaks observed at ~96 and ~110 cm$^{-1}$ originate from the E and A$_1$ modes of $\alpha$-IS[46], and the other two peaks located at ~132 and ~148 cm$^{-1}$ correspond to FGT[47]. The comparison between spectra obtained at 70 K (solid curves) and 300 K (dashed curve; black) clearly shows the temperature-dependent Raman peak positions moving towards lower wavenumbers (or redshift) as the temperature increases; for example, the strongest IS peak is observed at 103 cm$^{-1}$ at room temperature (RT; 300 K), such that the amount of redshift is ~7 cm$^{-1}$ per 230 K. This redshift of the Raman peak positions is mainly due to the lattice expansion of each material and consistent with previous reports[47–50].

The piezoelectric IS layer can generate strain in response to the varying $V_G$, which would result in changes to the Raman peak positions. In order to observe any $V_G$-dependent Raman peak position change, the $V_G$-dependent Raman spectra are fitted with the multiple-Voigt-function model; the IS and FGT peak positions are shown as dotted gray lines in Fig. 3a, and Fig. 3b summarizes the relation between the Raman shifts and $V_G$. For all the IS and FGT Raman peaks, we see the same trend with voltage. When $V_G$ is applied, the Raman peak positions of both IS and FGT decrease (redshift) compared to the zero bias case, regardless of polarity. This implies that the interfacial coupling between the IS and FGT layer is strong enough, such that the piezo-electric response of the IS layer is transferred to the ferromagnetic FGT layer. By analogy with the thermal-expansion-induced redshift (see previous paragraph), and from the fact that the in-plane lattice of IS expands as the out-of-plane polarization increases (opposite to conventional ferroelectrics)[35], the voltage-induced redshift of the Raman peaks corresponds to an in-plane tensile strain in the heterostructure. We estimate the tensile strain of IS to be 0.3–0.4 % at $V_G = 5$ V, based on the rate 3.1 cm$^{-1}$/% of Raman shift change with respect to strain, derived by Guo et al.[51] While the FGT Raman peak redshift suggests the presence of tensile strain in FGT, the exact amount of strain in the FGT cannot be estimated due to lack of earlier studies on Raman measurements of the strain state of FGT.

The polarization-induced surface charge modulation at the FGT/IS interface can once again be excluded by measuring the Raman spectra of the remanent states at zero bias after voltage application, which shows that the $V_G$-induced Raman peak redshift disappears when the bias is decreased back to zero (SI Fig. S9). Note that any remanent in-plane strain at zero bias during the voltage-sweep, due to the remanent ferroelectric polarization, should be minimal considering that the voltage-sweep range ($\pm 6$ V) is much larger than the expected ferroelectric coercivity ($E_c$) of the IS flake ($\pm 1$ V)[30]. While the increase of $E_c$ is expected at low temperatures, other well-known room temperature ferroelectric materials (e.g., BiFeO$_3$) show 30–100% increase of $E_c$ at cryogenic temperatures compared to room temperature[52], such that the voltages we applied for the MOKE and Raman measurements are likely larger than the $E_c$ of the IS flake even at 70 K. Also, note that the thermal effect due to Joule heating cannot explain the $V_G$-induced peak position change in Raman spectra since the FGT peaks are generally insensitive to temperatures below ~140 K[47]. Furthermore, in case the Raman peak position change is purely due to Joule heating, the amount of change in the peak positions we observe (i.e., around 1–2 cm$^{-1}$) would correspond to at least 30 K, assuming a linear relation between the data points at 70 and 300 K; this value is unreasonably large considering the small $I_G$ observed in the device. Therefore, we believe the $V_G$-induced tensile strain in the FGT/IS

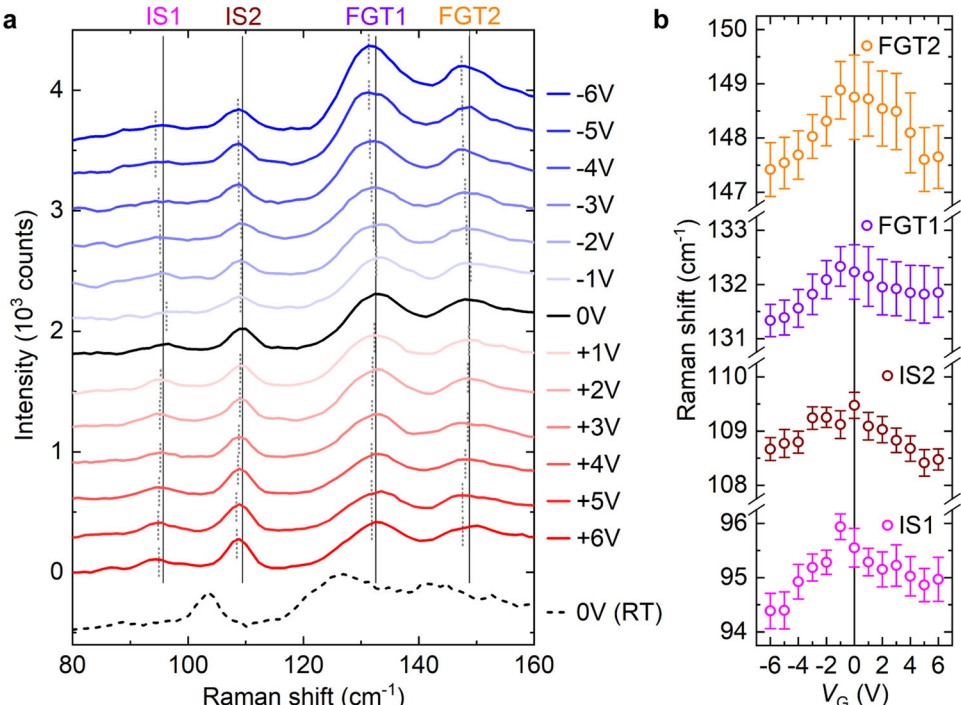

**Fig. 3 | Voltage-dependent Raman shifts. a** A series of Raman spectra of the FGT/IS heterostructure as a function of $V_G$, measured at 70 K. The measurements are performed as the magnitude of $V_G$ is progressively increased (0 V → +6 V, and 0 V → −6 V). After each $V_G$ application, Raman measurements are repeated at zero bias (see SI Fig. S9). The four fitted Raman peak positions at zero bias are shown as vertical solid black lines and the peak positions for each voltage application is shown as dotted gray lines. The dashed black line shows a Raman spectrum obtained at 300 K (RT) without any bias. Each plot is vertically spaced from neighboring plots by 300 counts. **b** A summary of $V_G$-dependent Raman peak positions. The error bars are defined in Methods.

heterostructure is the most likely origin of the consistent Raman peak redshifts that we observe.

From the experiments, it is clear that both the magnetic properties (especially, $H_c$) and the tensile strain in the FGT/IS heterostructure device show consistent voltage dependence. This suggests that the IS piezoelectric strain and accompanied change in lattice constants modulates the $H_c$ of the adjacent FGT. In general, $H_c$ of magnetic materials are determined by many material parameters, including the magnetic anisotropy, magnetization, magnetic domains, defects, etc. However, it is not likely that there will be voltage-induced changes in the defect states of the metallic FGT. We also argue that any voltage-dependent modulation of the magnetic domains would likely be a consequence of changes in the magnetic anisotropy[53]. Thus, it is reasonable to expect the observed reversible voltage-dependent change of $H_c$ can most likely be attributed to a voltage-controlled modulation of either the magnetization or the magnetic anisotropy[54].

### Theoretical calculations: strain-induced change in magnetic anisotropy

The mechanism of the strain-induced change in the magnetic properties of FGT (e.g., magnetization, magnetic anisotropy) is investigated by first-principles density functional theory (DFT), which reveals the existence of a substantial tensile-strain-induced decrease in the FGT perpendicular magnetocrystalline anisotropy. We assume 1 hole doping (h) per formula unit (f.u.) corresponding to the Fe-deficiency $x \approx 0.36$ mostly on the $Fe^{3+}$ site of $Fe_{3-x}GeTe_2$ (FGT) (see "Methods" section and ref. 28 for details). Figure 4a presents the magnetic moment per Fe and the magnetocrystalline anisotropy energy (MAE) with respect to the in-plane strain, where the MAE is defined as the total energy per Fe with in-plane ([210]) magnetization with respect to that with out-of-plane ([001]) magnetization; there is negligible energy change with rotation of the magnetization direction in the (001) plane. We find that there is almost no strain dependence in the magnetic

moment per Fe (Fig. 4a) and also find that the MAE monotonically decreases with increasing strain (Fig. 4a), showing a significant reduction in the MAE even for small strain states (0 ~ +0.4%). These suggest that the reduced FGT magnetic anisotropy, rather than any modulation of its magnetization, is the origin of the observed tensile-strain-induced reduction in $H_c$. We point out that the hole doping due to the Fe deficiency in our FGT specimen is crucial for this large strain-induced change in the MAE. That is, the hole doping decreases the overall MAE and drives the system in close proximity to the tensile strain-induced transition from out-of-plane to in-plane magnetocrystalline anisotropy. As the hole doping decreases, the MAE can become relatively insensitive to the strain, or in some cases show a tensile-strain-induced MAE increase (see SI Fig. S10). This doping-dependence on the MAE-strain relation suggests that hole doping is an additional parameter that can be utilized for subtle control of the magnitude and sign of the voltage-induced MAE modulation.

Further investigation of the strain-induced change in the electronic structures (for the 1 h per f.u. case, as discussed in the previous paragraph) reveals that the tensile strain induces the momentum-dependent band shifts. This results in the decrease (increase) in MAE around the Γ(K)-point, with the net effect being the decrease in the overall MAE. The momentum-resolved MAE $\xi_\mathbf{k}$ is defined as

$$\xi_\mathbf{k} = \frac{1}{N}\sum_n (f(\epsilon_{n\mathbf{k}}^{IP})\epsilon_{n\mathbf{k}}^{IP} - f(\epsilon_{n\mathbf{k}}^{OP})\epsilon_{n\mathbf{k}}^{OP}) \qquad (1)$$

where $\epsilon_{n\mathbf{k}}^{IP}$ and $\epsilon_{n\mathbf{k}}^{OP}$ are eigenvalues calculated with in-plane (IP) and out-of-plane (OP) magnetization, respectively, $N$ is the number of $k$-points, and $f(\epsilon)$ is the Fermi-Dirac distribution. Since $\xi_\mathbf{k}$ integrated over the Brillouin zone (BZ) comes to be the MAE, by analyzing the $\xi_\mathbf{k}$, we can identify $k$-points that mainly contribute to the MAE and also the momentum-dependent MAE changes with tensile strain. Figure 4b, c show the momentum-resolved MAE with 0% and 1% strain, and the

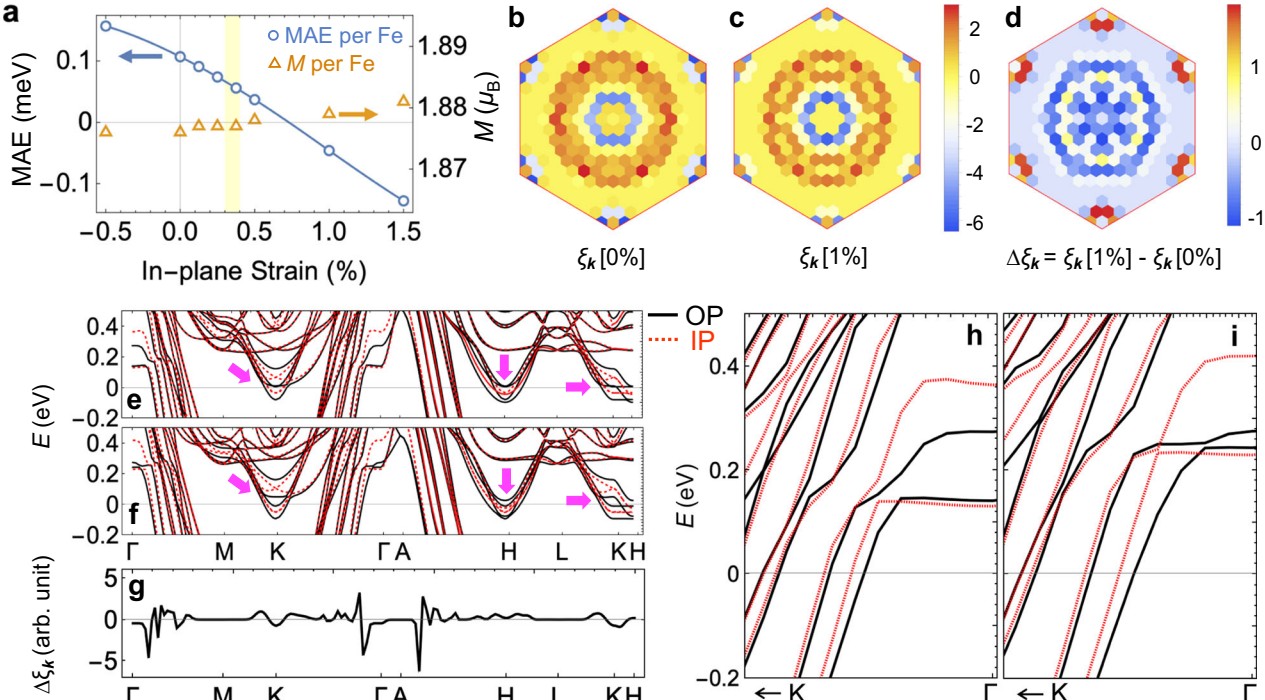

**Fig. 4 | Strain-dependent MAE and electronic band structures. a** MAE per Fe and magnetic moment per Fe as a function of in-plane strain. The yellow shaded region represents the estimated strain values of IS. The solid blue line is a guide to the eye. **b**, **c** Momentum-resolved MAE $\xi_{\boldsymbol{k}}$ (**b**) without strain and (**c**) with 1% tensile strain. The values of $\xi_{\boldsymbol{k}}$ are represented by the color scale on the right. **d** Tensile-strain-induced change in the momentum-resolved MAE relative to that of the unstrained case. **e**, **f** Band structures along the high symmetry lines (**e**) without strain and (**f**) with 1% tensile strain. Solid black (dotted red) lines show band structures with OP (IP) magnetization. **g** Tensile-strain-induced change in the MAE along the high symmetry lines. **h**, **i** Band structures near the Γ-point along the Γ-K direction (**h**) without strain and (**i**) with 1% tensile strain.

difference between the two is depicted in Fig. 4d. We choose results with 0% and 1% tensile strain to depict the strain dependence more clearly. Nevertheless, the general trend is the same for smaller strain differences, consistent with the monotonic strain dependence shown in Fig. 4a. The $\xi_{\boldsymbol{k}}$ areshown in the in-plane BZ, where the out-of-plane momentum ($k_z$) contribution of $\xi_{k+k_z}$ are combined for each in-plane crystal momentum $k$. We find that for both unstrained and strained cases, the $\xi_{\boldsymbol{k}}$ shows a similar $k$-dependent pattern. As the $k$-vector moves away from the Γ-point, there is an alternating contribution to MAE, with negative (positive) values for small (large) magnitude of $k$-vectors. In the vicinity of the K-point, there is also substantial contribution to MAE, with positive MAE contribution in close proximity to the K-point and negative MAE further away from the K-point. When all the contributions in the in-plane BZ are summed, the overall MAE changes sign from positive (perpendicular anisotropy) to negative (in-plane anisotropy) with a large 1% tensile strain as shown in Fig. 4a. The difference in the $k$-dependent MAE between the unstrained and strained cases reveals that the negative MAE contributions around the Γ-point dominate, resulting in a net negative change in MAE.

Comparison of the band structures and the associated change in the momentum-dependent MAE reveals how the strain-induced band shifts result in the large strain-induced modulation of the total MAE. Figure 4e, f show band structures around the Fermi energy with in-plane magnetization (dotted red lines) and out-of-plane magnetization (solid black lines) with 0% and 1% tensile strain. Given the large positive eigenvalue ~8.5 eV for $\varepsilon_{n\mathbf{k}}$ around the Fermi energy, Eq. (1) suggest that the positive MAE at a $k$-point is expected when the number of occupied states with in-plane magnetization is larger than that with out-of-plane magnetization ($\epsilon_{n\mathbf{k}}^{\text{IP}} < \epsilon_{\text{F}} < \epsilon_{n\mathbf{k}}^{\text{OP}}$) and negative MAE for vice versa ($\epsilon_{n\mathbf{k}}^{\text{OP}} < \epsilon_{\text{F}} < \epsilon_{n\mathbf{k}}^{\text{IP}}$).

We find that there are two major changes in the band structures in the presence of tensile strain. First, there is splitting of the $\epsilon_{n\mathbf{k}}^{\text{OP}}$ bands

(marked with magenta arrows), which increase the occupancy of the out-of-plane bands right at the K-points, but decrease the occupancy at $k$-points around the K- and H-points. Since the IP bands are relatively insensitive to the strain, this gives an overall positive MAE around the K- and H-points and small negative MAE right at the K-point as shown in Fig. 4d, g. Second, there are large upshifts of the bands around the Γ-point. Figures 4h and 4i show a magnified view of the electronic bands near the Γ-point, where the band upshift effect (i.e., increased splitting between $\epsilon_{n\mathbf{k}}^{\text{OP}}$ and $\epsilon_{n\mathbf{k}}^{\text{IP}}$) is most clearly seen. Since $\epsilon_{n\mathbf{k}}^{\text{OP}} < \epsilon_{\text{F}} < \epsilon_{n\mathbf{k}}^{\text{IP}}$, the increased splitting contributes to the negative MAE around the Γ-and A-points (Fig. 4d, g). We note that the reduction in MAE from the shift of bands with large $\epsilon_{n\mathbf{k}}^{\text{IP}}$ and $\epsilon_{n\mathbf{k}}^{\text{OP}}$ band splitting away from the Fermi energy is consistent with the doping-induced reduction of MAE[28]. The strain-dependent analysis of the FGT band structure confirms that tensile strain can indeed induce significant modulation in its MAE, supporting our experimental results.

## Discussion: role of the interfacial polymer layer in the strain-mediated magnetoelectric effect

We note that the overall homopolar voltage effect has some temperature-dependent and sample-to-sample variations (Figs. 2a–e, and SI S2). At 67 K, the voltage-dependent $H_c$ modulation shows some polarity-dependent asymmetry, in contrast to the more or less symmetric voltage-dependent $H_c$ modulation at 80 K. Additionally, the overall voltage-dependent $H_c$ decrease shows some minor sample-to-sample variations (see SI Fig. S2). We suggest the properties of the polymer layer existing at the FGT/IS interface as the possible origin of these temperature-dependent and sample-to-sample variations in the detailed features of the voltage-induced $H_c$ modulation. As discussed earlier, we believe the interfacial polymer at the FGT/IS interface acts as a flexible adhesive layer to transfer strain between the two vdW materials. Yet, there could be small variations in the amount of strain transfer

for each voltage-induced-strain-application owing to its plastic properties, resulting in minor differences in the voltage dependence.

Full validation of the proposed strain-mediated mechanism requires further investigation, and we propose follow-up studies. A comparison between samples with and without interfacial polymer layers can reveal the exact role of the thin interfacial polymer layer. However, the removal of this interfacial layer has been difficult (see "Methods" section). Nevertheless, we obtain consistently similar polymer thickness under our sample fabrication conditions (see SI Fig. S11). Here, we take advantage of the polymer-mediated-strain-transfer to implement voltage-controlled magnetoelectric effect in a vdW heterostructure, and emphasize that a consistent voltage effect, i.e., the homopolar voltage-induced $H_c$ decrease, is always observed in all the FGT/IS heterostructure devices studied.

In fact, in an earlier study[55], the presence of the inverse magnetostriction effect in $Fe_3GeTe_2$ was revealed by probing the strain-dependent modulation of the magnetic properties of $Fe_3GeTe_2$ directly transferred on a flexible polyimide substrate. A large increase in the $H_c$ of $Fe_3GeTe_2$ was reported when an in-plane tensile strain was applied to the polyimide substrate, confirming the existence of the inverse magnetostriction effect in $Fe_3GeTe_2$. The tensile-strain-induced $H_c$ decrease that we observe in our work seems opposite to the tensile-strain-induced $H_c$ increase in the study. However, both of these seemingly contradictory tendencies can in fact be well understood by our DFT calculation results. In SI Fig. S10, the MAE decreases with tensile strain for the 1h-doped FGT ($Fe_{2.64}GeTe_2$) we use in our work, but the MAE increases with strain for lower-level-hole-doping cases in agreement with the tensile-strain-induced $H_c$ increase in the un-doped $Fe_3GeTe_2$ observed by Yu Wang et al.[55]

In conclusion, we investigate the voltage control of magnetism in a vdW ferromagnetic/ferroelectric FGT/IS heterostructure device. Voltage-dependent MOKE measurements on the heterostructure device show a decrease in FGT coercive field for both positive and negative biases, while voltage-dependent Raman spectroscopy reveals the existence of voltage-dependent tensile strain states of FGT and IS, regardless of voltage polarity. The consistent voltage-dependent change in both the magnetic properties and strain state suggests that the voltage-induced piezoelectric strain in IS gives rise to changes in the magnetic properties of the adjacent ferromagnetic FGT. Density functional theory calculations show a large strain-dependent change in the FGT magnetic anisotropy energy, confirming the coupling between strain and magnetism in a vdW ferromagnetic/ferroelectric heterostructure.

## Methods

### Single crystal growth
Single crystals of $Fe_{2.64}GeTe_2$ (FGT) are grown from molten metallic fluxes, as reported previously[25]. Single crystals of graphite, h-BN, and 2H $\alpha$-$In_2Se_3$ (IS) are commercially available from HQ Graphene.

### Heterostructure device fabrication
The substrate is a 5 mm × 5 mm size Si wafer with a 300-nm-thick thermally oxidized $SiO_x$ layer. Au (100 nm)/Ti (5 nm) electrodes are deposited by electron-beam evaporation after photo-lithography with AZ4210 polymer coated and baked (110 °C, 90 seconds) on the substrate. Graphite nanoflakes are exfoliated from single crystals with scotch tape onto polydimethylsiloxane (PDMS) blocks, and then transferred onto Au/Ti electrodes on the substrate, which allows the Au/Ti electrodes to become closer in lateral distance. IS and FGT nanoflakes are exfoliated and transferred onto the substrate by the same dry transfer method inside an Ar-filled glovebox ($H_2O$: <0.8 ppm; $O_2$: <0.7 ppm) to prevent sample oxidation and degradation during fabrication. The completed device consists of a FGT/IS/graphite heterostructure in which a 12-nm-thick graphite (bottom layer) and a 10.5-nm-thick (13 layers) FGT (top layer) are spaced by a 50-nm-thick IS. The FGT is connected to the Au/Ti electrode through 5-nm-thick graphite.

A large 20-nm-thick h-BN flake fully covers the FGT in order to minimize any chemical reaction or oxidation of FGT in air.

In addition to PDMS, we use various types of polymer materials (PC, PPC, gel-pak) for this dry transfer method, and as discussed in the main text, we always find the presence of an interfacial polymer layer. Typically, releasing the vdW material cleanly from the polymer requires a heating process. The difficulty lies in that the heating process should be limited when handling FGT; the ferromagnetic properties of FGT disappear after high-temperature (>80 °C) annealing. Additionally, applying chemical or etching methods, which alters the ferroelectric and ferromagnetic properties of the IS and FGT, respectively, should be avoided. Due to these restrictions, we are not able to prepare polymer-free interfaces with the dry transfer method. Nevertheless, the voltage effect on the FGT magnetic properties is consistently observed regardless of the polymer material used for the fabrication process (see SI Fig. S11 for samples fabricated by the dry transfer method using gel-pak).

### TEM measurement
The TEM specimens are prepared by a focused ion beam (Hitachi; NX5000) using Ga and Ar ions. The high-angle annular dark-field (HAADF) STEM images are acquired using monochromated and Cs-corrected TEM (FEI;Titan S80 – 300) equipped with a Gatan energy filter (GATAN; GIF Quantum ERS System, model 966). The energy dispersive spectroscopy (EDS) spectra and maps are obtained from multiple EDS detector TEMs (FEI; Talos F200X).

### Magnetic measurements
Magnetic hysteresis loops of FGT are measured at cryogenic temperatures by a polar (out-of-plane) MOKE measurement system built by Neoark Corporation. The MOKE system has Kerr rotation detection sensitivity <0.1 mrad, and utilizes a polarized 408 nm diode laser. The measuring laser beam spot size (FWHM) is ~2 μm on the sample, but due to the Gaussian shape of the beam, signals can be detected >5 μm away from the center. Considering the small lateral size of the FGT/IS heterostructure (5–10 μm), the obtained Kerr rotation is the magnetic signal averaged over a majority of the heterostructure device. Yet, it is possible that some part of the Gaussian-shaped laser beam can drift outside of the heterostructure, i.e., where there is no FGT layer, in the presence of inevitable random sample drifts, which can result in some variation in the Kerr rotation signals. The MOKE data are averaged over 3–10 measurements in order to compensate, to some degree, for this measurement-to-measurement variation issue. The gate voltage $V_G$ is always applied through the graphite under IS while FGT is grounded (Fig. 1b).

### Raman spectroscopy measurements
Raman measurements are conducted at 300 and 70 K with a micro-Raman spectrometer with 1200 g mm$^{-1}$ (Princeton Instruments; HRS-500) and a cryogenically cooled Si CCD camera (Princeton Instruments; PyLoN-100BRX) at the Multidimensional Materials Research Center (MMRC) of Kyung Hee University. A 514.4-nm CW laser (Cobolt; Fandango), whose bandwidth is further narrowed using Optigrate's Bragg grating bandpass and notch filters for low energy Raman measurement, is attenuated with one of the selected neutral density (ND) filters and focused down to a 2 μm spot on the samples. The spectral resolution is 1 cm$^{-1}$. The Raman peak positions are determined by fitting the measured data using the multiple-Voigt-function-model[43,56]. Multiple peaks are simultaneously fitted and the best-fit parameters (positions, heights, and widths) are found. The peak positions determined from the best-fit are plotted in Fig. 3b and Fig. S9b. In our experiments, we observe small fluctuations in the Si Raman peak positions; this could be due to simple measurement errors, or small changes in the calibration. This implies that there will be inevitable fluctuations in the IS or FGT Raman peak positions, which we define as

the error in determining the IS or FGT Raman peak positions. Thus, using the measured Si peak fluctuation, Si peak FWHM, and FGT or IS peak FWHM, we can estimate

$$\text{Error bar} = (\text{FGT or IS peak fluctuation}) = (\text{Si peak fluctuation}) \times \frac{(\text{FGT or IS peak FWHM})}{(\text{Si peak FWHM})}$$

The gate voltage $V_G$ is always applied through the graphite under IS while FGT is grounded (Fig. 1b).

## First-principles calculations

First-principles DFT calculations are performed using the Vienna ab-initio simulation package (VASP)[57,58]. The generalized gradient approximation (GGA) is used for the exchange-correlation functional with the Perdew-Burke-Ernzerhof (PBE) parametrization[59]. The projector augmented wave method[60] is used with the energy cutoff of 600 eV and Γ-point centered $16 \times 16 \times 5$ k-point grid. Spin–orbit coupling is included. The MAE is calculated using the force theorem[61,62]. Hole-doping is treated by reducing the total number of electrons with compensating uniform background charge. For each hole doping case, the atomic structures are obtained by relaxing the internal atomic coordinates with inclusion of the vdW interaction using the DFT-D2 method of Grimme[63] where the experimental lattice constants with Fe-deficiency (corresponding to hole doping) is used. When there are no corresponding experimental values, the lattice constants are obtained by interpolating the known the reported experimental values. The electronic structures with in-plane strain are obtained by strained-bulk calculation where the in-plane lattice constants are changed according to the strain and the out-of-plane lattice constant is determined such that the unit-cell volume is conserved. With the fixed lattice constants, the internal-coordinate relaxation is performed.

## Data availability

All data generated in this study are provided in the article and Supplementary Information. Additional data and materials are available from the corresponding authors upon request.

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

## Acknowledgements

J.W.C., J.H.L., J.E., H.R., and C.J. were supported by the KIST Institutional Program (2E32251, 2E32252), and the National Research Foundation of Korea (NRF) grants funded by the Ministry of Science and ICT (MSIT) (NRF-2021R1A2C2011007, NRF-2021M3H4A1A03054856, NRF-2020R1A5A1016518, and NRF-2021R1A2C2014179). S.Y.P. was supported by the NRF grant funded by the MSIT (No. 2021R1C1C1009494) and by Basic Science Research Program through the NRF funded by the Ministry of Education (No. 2021R1A6A1A03043957). C.-H.Y. acknowledges the support of the Korean Government via the Creative Research Initiative Center for Lattice Defectronics (No. 2017R1A3B1023686). Materials synthesis by S.C. and C.P. was supported by the Office of Basic Energy Sciences, Materials Sciences and Engineering Division, U.S. DOE under Contract No. DE-SC0012704. M.C and Y.D.K was supported by the NRF grant funded the MSIT (2022R1A4A3030766, 2021R1A2C2093155). Y.D.K acknowledges MMRC of Kyung Hee University (Grant No. 2021R1A6C101A437).

## Author contributions

J.W.C. conceived the project on voltage control of magnetism in vdW heterostructures. J.H.L. designed the strain analysis investigation. S.Y.P. led and conducted the theoretical calculations and analysis. J.E. fabricated the heterostructure device with assistance from I.L. and J.Y.K., and conducted the magnetic measurements. J.E. and M.C. carried out the Raman spectroscopy measurements. J.H.L, J.E., Y.S., Y.D.K., and M.-J.S. carried out the Raman analysis. H.S. and H.J.C. conducted the TEM measurements. J.S., H.-J.C., S.-H.B., and C.-H.Y. characterized the ferroelectric properties of the device. S.C. and C.P. synthesized the FGT single crystals. J.E., J.H.L., S.P., and J.W.C wrote the manuscript, with assistance from H.R., C.J., Y.D.K., M.-J.S., and comments from all the authors. J.H.L., S.P., and J.W.C. supervised the work.

## Competing interests

The authors declare no competing interests.
