## [Peer Review File · Nature Communications]

Voltage control of magnetism in Fe_{3-x}GeTe₂/In₂Se₃ van der Waals ferromagnetic/ferroelectric heterostructuresReviewers' comments:

Reviewer #1 (Remarks to the Author):

In this manuscript, Eom et al studied the magnetoelectric coupling in a vdW Fe_{3-x}GeTe₂/In₂Se₃ heterointerface. In their device, they found that the gate voltage applied to the Fe_{3-x}GeTe₂/In₂Se₃ heterostructure device can modulate the magnetic properties of Fe_{3-x}GeTe₂ with a large decrease of coercive field. Combined with Raman spectroscopy and theoretical analysis, they attribute the large tune of coercive field in FGT to the in-plane tensile strain induced by piezoelectric response of In₂Se₃ layer under gate voltages. This experiment stands in the continuation of efforts of van der Waals ferromagnetic heterostructure for spintronic devices, it may also provide another knob to control the magnetism via the tensile strain. Having said that, however, I find the current data does not support the conclusions, thus I cannot recommend this article for publication at least in present form. More reproducible experimental results are required to support the claims.

1. Is the observed magnetoelectric coupling effect reproducible? I find their main experimental data is obtained from a single device (10.5 nm FGT/50 nm IS/Graphite), the authors should repeat the experiment in at least another heterointerface to confirm their observations.

2. In this experiment, the magnetoelectric coupling effect comes from the heterointerface, this effect should be thickness (FGT layer) dependent. Did the authors observe any enhancement (or attenuation) of magnetoelectric coupling effect in thinner (or thicker) FGT layer?

3. Regarding their device fabrication, how did the authors transfer the IS and FGT layers? Did they transfer the FGT and IS layer one by one via PDMS or use FGT layer to pick up IS layer first and then transfer the IS/FGT bilayer onto graphite? What's the chemical composition of polymer layer at FGT/IS interface? Is it PDMS?

4. In Fig. 3a, there is indeed a Raman shift in negative voltage region (FGT1 peak), however, it seems the FGT1 peak doesn't shift in the positive voltage region, this is contradictory to their statement that coercive field decreases in both positive and negative voltages. The authors should clarify this point.

5. In the main text, the authors claimed that the tensile strain in IS induced by a gate voltage of 5 V is around 0.3-0.4%, however, the in-plane strain transferred onto the FGT layer is as large as 0.5 %. Generally, a large transfer of in-plane strain along out-of-plane direction is very challenging, the author should be careful regarding the estimation of the tensile strain in FGT layer.

Reviewer #2 (Remarks to the Author):

In this manuscript, the authors experimentally demonstrate magnetoelectric interactions in a Van der Waals (vdW)-a material-based device consisting of a ferromagnetic (Fe_{3-x}GeTe₂, x≈0.36) and a ferroelectric (α -In₂Se₃) vdW material. The significant decrease in the coercive field for both positive and negative bias voltages is attributed to the tensile strain-induced reduction of the perpendicular magnetocrystalline anisotropy (MCA), supported by density

functional theory calculations. The strain change is from the voltage-induced piezoelectric response on In₂Se₃. The results show an interesting and important topic of low-power voltage-controlled vdW spintronic devices. Nevertheless, several experiment data may conflict with their conclusion. Thereby, the author should answer the following comments before further consideration.

1. From Figs 2a, 2b, and S3, the reversible manner of FGT/IS can be observed by applying and withdrawing the voltages. V_g-induced H_c modulation seems reversible and mostly non-remanent within one cycle. The reversibility should be investigated at least 20 cycles to show its stable and reversible manner.

2. I suggest the author replenish H_c curves by applying the -4.5 V and -5 V in Fig 2b during negative V_G-induced H_c variation. It is helpful for the reader to understand better.

3. The authors declare that the IS surface charge effect can be ruled out, including the current or spin torque effect. Thus, the author needs to give a mechanism explanation about H_c variation issues. For example, from Figs S3, the H_c becomes narrow with approximately the same tendency and shape in FGT under -5 V and 5 V at 80 K, respectively. However, this phenomenon can not be obtained in FGT under -5 V and 5 V at 67 K. What happened? On the other hand, the saturation moments are altered after the reversible test. Why? The magnetic moments should be back to the original position during the manipulation of the ferromagnetism of FGT. Is there another factor left to exclude?

4. Van der Waals (vdW) materials are strongly bonded two-dimensional (2D) layers bound in the third dimension through weaker dispersion forces. Compared with thin-film-based composite systems, dispersion forces are much smaller than chemical bonds. How to effectively couple the electric and magnetic properties at the ferromagnetic/ferroelectric vdW interface?

5. The authors claim that "Second, the effect of surface charge induced by the out-of-plane polarization of IS can be excluded since H_c decreases for both positive and negative voltages." However, the non-symmetric voltage-induced H_c decrease in Figure 2 (c) is inconsistent with the symmetric voltage-induced tensile strain in Figure 3 (b). The polarization-induced surface charge modulation should be considered.

6. Also, deliberately adding a charge dissipation layer underneath the FGT would help to rule out polarization-induced surface charge modulation.

7. From MAE to H_c, more debates should be added. In addition, the decreased saturation magnetization also will lower the H_c.

8. For some details, the author may condense the abstract. Also, the Scale bar should mark its magnitude in Figures 1 and S1; Add some recent articles in the Introduction.

Reviewer #3 (Remarks to the Author):

Voltage control of magnetism is important for developing low-power spintronic devices. Much progress have been done in ferromagnetic/ferroelectric multiferroic heterostructures owing to strain-mediated magnetoelectric coupling. Moreover, the control of magnetism in

ferromagnetic/ferroelectric vdW heterostructures has been predicted in many previous papers. This paper is an experimental attempt to realize this aim. Although the change of coercive field by applying voltages is observed, however, the evidence supporting the conclusion is insufficient.

1. The coercive field is $\sim 25\text{mT}$ at 67 K, but the typical H_c for FGT ($\sim 10\text{nm}$) is 2500 Oe (250mT). [Tan, Cheng, et al. Nature communications 9, 1554 (2018).
2. The mechanism of decrease in coercive field is questionable. More control experiments should be performed. For example, use non-ferroelectric $\beta\text{-In}_2\text{Se}_3$ to replace the ferroelectric $\alpha\text{-In}_2\text{Se}_3$ and perform the same measurements, given that $\alpha\text{-In}_2\text{Se}_3$ and $\beta\text{-In}_2\text{Se}_3$ have similar semiconducting property. The author performed a contrast experiment on FGT/h-BN, but this is not a valid contrast because h-BN is a wide-bandgap insulator, not comparable with $\alpha\text{-In}_2\text{Se}_3$ in terms of electric properties.
3. The biggest problem is that the author make comparison between data acquired from nonequilibrium state and equilibrium state. Such comparison is not reliable. In the nonequilibrium state where V_g and I_g is not zero, many side effects, not only current-induced effects, but also other spin-orbit coupling effects, can occur, which make the spin system different from that in equilibrium state. I would suggest to insert a thin hBN layer between graphite and In_2Se_3 . This would effectively eliminate I_g but maintain the exertion of voltage.
4. The author state that current-induced effects are eliminated, but based on doubtful evidence, which is not convincing enough: (1) I_g is 4 times larger for negative bias than that of positive bias, but H_c seems independent from the voltage polarity. This is a very weak support for the claim. As mentioned by the author in the beginning part, there are some polymer residues on the interfaces, therefore, the current may not distribute evenly on the interfaces. The MOKE measures the magnetic properties only on a small point, not the whole junction. The current density on the measured point may not be 4 times larger as the author announced. (2) The author states that there is no electrically induced H_c change in FGT without ferroelectric. This is not a convincing reason. FGT forms barrier-free Ohmic contact with metal electrodes, whereas in FGT/ In_2Se_3 heterostructure, as shown by Figure 1c, Schottky barrier exists, resulting in possible unevenly distributed heating, making the heating more prominent and not negligible.
5. More importantly, considering the ferroelectric polarization of In_2Se_3 is permanent, the voltage control effect is supposed to be nonvolatile. The non-remanent phenomena the author reported is conflict with this intuitive reasoning. [Huang, Xiaokun, et al. Physical Review B 100.23 (2019): 235445.]
6. In Line 201, it is mentioned that ferroelectric switching voltage is $<1\text{ V}$ for 50-nm-thick In_2Se_3 . However, the author should be aware that this is only applicable for the sample at room temperature. For low temperature (67 K in this work), the situation can be very different, due to the reduced domain kinetics, suppressed thermal assisted switching, et al.
7. In line 165, "67K" should be "67 K".
8. What is the T_c of FGT? Does the applied voltage have any effect on the T_c ?
9. From Fig. 1c, the current is about $-40\ \mu\text{A}$ at -5 V , indicating a power consumption of $200\ \mu\text{W}$. When measuring the MH loops, the voltage is always applied so that a significant Joule heating effect can not be avoided, which will increase the temperature. Please give the comment on effect of Joule effect.
10. Voltage dependent H_c is asymmetric at 67 K (Fig. 2c) and symmetric at 80 K (Fig. S2). Can author explain the origin of this difference?
11. For strain-mediated magnetoelectric coupling in ferromagnetic/ferroelectric multiferroic heterostructures, the magnetism is modulated by piezostrain through inverse magnetostriction effect. What is the magnetostriction coefficient of FGT?
12. In Fig. 3a and Fig. S5a, it is hard to find the Raman peak. You'd better plot the Raman

peaks at various voltages with a curved line rather than a straight line. How do you get the error bar in Fig. 3b and Fig. S5b? what is the resolution of Raman spectroscopy?

Response to the reviewer's comments

We deeply appreciate the reviewers' time and effort in evaluating our manuscript. We have performed a major revision of our manuscript based on the reviewers' constructive suggestions, which has significantly improved the quality of this work. The revised manuscript has become scientifically more rigorous, specifically by including additional experimental data and further analysis & discussion on the experimental results. Hence, we strongly believe that the revised manuscript is suitable for publication in *Nature Communications*. Below we address the comments in detail.

Reply to Reviewer #1

Remarks to the Author:

In this manuscript, Eom et al studied the magnetoelectric coupling in a vdW $Fe_{3-x}GeTe_2/In_2Se_3$ heterointerface. In their device, they found that the gate voltage applied to the $Fe_{3-x}GeTe_2/In_2Se_3$ heterostructure device can modulate the magnetic properties of $Fe_{3-x}GeTe_2$ with a large decrease of coercive field. Combined with Raman spectroscopy and theoretical analysis, they attribute the large tune of coercive field in FGT to the in-plane tensile strain induced by piezoelectric response of In_2Se_3 layer under gate voltages. This experiment stands in the continuation of efforts of van der Waals ferromagnetic heterostructure for spintronic devices, it may also provide another knob to control the magnetism via the tensile strain. Having said that, however, I find the current data does not support the conclusions, thus I cannot recommend this article for publication at least in present form. More reproducible experimental results are required to support the claims.

Reply: We deeply thank the reviewer for the careful reading and critical comments. As the reviewer points out, our experiments demonstrate an effective method to realize voltage control of magnetism in vdW spintronic devices. We believe our observation of the magnetoelectric effect in a van der Waals (vdW) ferromagnetic-ferroelectric heterostructure provides a significant advance towards the realization of low-power voltage-controlled vdW/2D spintronic devices.

Following the constructive suggestions of the reviewer, we have conducted additional experiments to support and validate our claim. In the revised manuscript, we include additional data, along with further analysis. Below, we provide a point-by-point response to the comments and questions, along with the list of necessary revisions made to the manuscript.

1. Is the observed magnetoelectric coupling effect reproducible? I find their main experimental data is obtained from a single device (10.5 nm FGT/50 nm IS/Graphite). The authors should repeat the experiment in at least another heterointerface to confirm their observations.

Reply: We completely agree with the reviewer that we should be able to reproduce the observed effect in other FGT/IS heterostructure devices in order to claim the existence of magnetoelectric coupling in FGT/IS. Following the suggestion of the reviewer, we repeated the experiments in three other independent FGT/IS heterostructure devices. As seen in Fig. R1 (included in revised Supplementary Information as Fig. S2), we find that the voltage-induced modulation of the magnetic coercivity is consistently observed in all the FGT/IS heterostructure devices studied in this work. Despite some sample-to-sample variations, the overall tendency remains the same: the FGT magnetic coercivity decreases with increasing gate voltage, regardless of voltage polarity.

Fig. R1 (SI Fig. S2). V_G -dependent H_c modulation in other FGT/IS devices.

a) Optical microscopy (OM) image of the FGT(12 nm)/IS(100 nm) device. b) V_G -dependent M - H loops and c) H_c plotted as a function of V_G for the device shown in (a). Measurements at 70 K.

d) OM image of the FGT(10 nm)/IS(40 nm) device. e) V_G -dependent M - H loops and f) H_c plotted as a

function of V_G for the device shown in (d). Measurements at 74 K.

g) OM image of the FGT(20 nm)/IS(50 nm) device. h) V_G -dependent M - H loops and i) H_c plotted as a function of V_G for the device shown in (g). Measurements at 80 K.

List of Changes (highlighted yellow in manuscript):

- Included discussion in the main text (page 7):

“This V_G -dependent H_c decrease is consistently observed in other FGT/IS devices (SI Fig. S2).”

- Included Fig. R1 (see above) as Fig. S2 and caption in SI (page 4).

2. In this experiment, the magnetoelectric coupling effect comes from the heterointerface, this effect should be thickness (FGT layer) dependent. Did the authors observe any enhancement (or attenuation) of magnetoelectric coupling effect in thinner (or thicker) FGT layer?

Reply: We thank the reviewer for this critical comment. We agree that in the case the polarization-induced IS surface charge is the origin of the V_G -induced H_c modulation, the magnetoelectric effect would show a significant FM-layer-thickness-dependence. However, in our case of piezo-strain-mediated magnetoelectric coupling, the voltage effect can show a small FM-layer-thickness-dependence as long as the piezoelectric in-plane strain can be uniformly transferred from the FE to FM layers (Please refer to our reply to comment 5 on more discussion of the polymer-mediated in-plane strain transfer mechanism); we find that in the range of sample thicknesses that we have tried (up to 20 nm), there is no significant thickness dependence on H_c .

In Fig. R2 below, we plot the FGT-thickness-dependent H_c modulation. The H_c is plotted as a function of E_G ($\equiv V_G / \text{IS-thickness}$) for the FGT/IS devices studied. We find that the voltage effect does not show significant FGT-thickness-dependence in the thickness range 10~20 nm. This implies that the polymer-mediated in-plane strain transfer is effective up to ~20-nm-thick FGT layers. Note that in all the heterostructure devices studied, FGT thickness (10~20 nm) \ll IS thickness (50~100 nm).

In order to systematically investigate and confirm the interfacial nature of the voltage effect, a more thorough study involving FGT layers in a wider thickness range is required. However, for FGT thickness > 30 nm, the magnetic stripe domain phase appear [Nano Lett. 18, 5974 (2018); Nano Lett. 20, 95 (2020)], with characteristic split M - H loops which makes it difficult to define H_c , and hence, only thin (< 20 nm) FGT layers are used for this study.

Fig. R2 (SI Fig. S3). FGT-thickness-dependent H_c modulation. H_c plotted as a function of E_G ($\equiv V_G / \text{IS-thickness}$) for the FGT/IS devices shown in Fig. 1b, Fig. S2a, Fig. S2g. The H_c values are normalized to the H_c at $E_G = 0$ V/nm for each heterostructure device. The labels indicate the FGT thickness and measurement temperature.

List of Changes (highlighted yellow in manuscript):

- Included discussion in the main text (page 7):

“We also find that the voltage effect does not show FGT-thickness-dependence up to 20 nm (See SI Fig. S3 and discussion).”

- Included Fig. R2 (see above) as Fig. S3 and caption in SI (page 5).

- Included discussion following Fig. S3 in SI (page 5):

“The voltage effect does not show any significant FGT-thickness-dependence in the thickness range 10.5~20 nm. This implies that the voltage-induced-piezo-strain transfer between the IS and FGT is effective up to ~20-nm-thick FGT layers. Note that in all the heterostructure devices studied, FGT thickness (10.5~20 nm) \ll IS thickness (50~100 nm). In the case the polarization-induced IS surface charge is the origin of the V_G -induced H_c modulation, the effect would show a more obvious FGT-thickness-dependence. For FGT thickness > 30 nm, the magnetic stripe domain phase appear^{3,4}, with characteristic split M - H loops which makes it difficult to define H_c , and hence, only thin (< 20 nm) FGT layers are used for this study.”

- Included references 3 and 4 in SI (page 13):

3. Regarding their device fabrication, how did the authors transfer the IS and FGT layers? Did they transfer the FGT and IS layer one by one via PDMS or use FGT layer to pick up IS layer first and then transfer the IS/FGT bilayer onto graphite? What's the chemical composition of polymer layer at FGT/IS interface? Is it PDMS?

Reply: We thank the reviewer for pointing out the ambiguity of our description about transfer of the vdW materials and the polymer layer in between. We exfoliate graphite, IS, FGT, and hBN nanoflakes from their respective single crystals onto PDMS blocks on slide glasses, and then transfer them onto a substrate *one by one*. We do not use FGT flakes to pick up IS. As a result, PDMS makes contact with the top sides of every flake transferred onto the substrate, leaving some polymer residue on them. In the cross-sectional TEM images and EDS graphs of FGT/IS heterostructure devices (See the SI Figure S1), we find that the polymer layer is chemically composed of C, Si, and O, all of which are also the full chemical composition of PDMS. Hence, we suspect it is the residue from the vdW material transfer process using PDMS.

List of Changes (highlighted yellow in manuscript and SI):

- Included discussion in the main text (page 19):

“IS and FGT nanoflakes are exfoliated and transferred one by one onto the substrate by the same dry transfer method with PDMS”

- Included discussion in the SI (page 3):

“The interfacial layer has detectable quantities of carbon, silicon, and oxygen, all of which are also the full composition of PDMS. It may look as if the interfacial layer is mixed with FGT and IS according to the EDS graph (Fig. S1c), but judging from the clear boundaries of each layer (Fig. S1a), we believe it results from the relatively low resolution of the EDS measurement. Therefore, we suspect it is the residue from the vdW material transfer process using PDMS.”

4. In Fig. 3a, there is indeed a Raman shift in negative voltage region (FGT1 peak), however, it seems the FGT1 peak doesn't shift in the positive voltage region, this is contradictory to their statement that coercive field decreases in both positive and negative voltages. The authors should clarify this point.

Reply: We thank the reviewer for examining our data attentively. As the reviewer points out, the FGT1 Raman peaks show a clear redshift in the negative voltage region, but the redshift in the positive region

seems less obvious in our original Fig. 3a. The FGT Raman peak positions are obscured by the numerous Raman peaks which overlap in the measuring range (80 to 160 cm^{-1}). In order to quantify the FGT1 and FGT2 Raman peak positions, we fit the Raman spectra using the multiple-Voigt-function model. In Fig. R3, we plot the fitted peaks of the voltage-dependent FGT1 and FGT2 Raman spectra (see dotted curves in Fig. R3 below). We observe a small yet noticeable redshift of the FGT1 Raman peak at positive voltages. In the revised Fig. 3a, we plot the IS1, IS2, FGT1, and FGT2 Raman peak positions at different V_G 's as gray dotted lines and the Raman peak positions at zero bias as black solid lines, such that the voltage-dependent Raman peak shifts can be better visualized. We also note that this result is insensitive to the fitting functions; the same voltage-dependent redshift is also observed when the Raman spectra are fitted using the pseudo-Voigt functions.

Fig. R3. The fitted curves (using multiple Voigt functions) of the FGT1 and FGT2 Raman peaks.

List of Changes (highlighted yellow in manuscript):

- Included discussion in the main text (page 11):

“In order to observe any V_G -dependent Raman peak position change, the V_G -dependent Raman spectra are fitted with the multiple-Voigt-function model; the IS and FGT peak positions are shown as gray dotted lines in Fig. 3a,”

- Revised Fig. 3a and caption in main text (page 13):

Fig. 3a. The four fitted Raman peak positions at zero bias are shown as solid black lines and the peak positions for each voltage application is shown as gray dotted lines.

5. In the main text, the authors claimed that the tensile strain in IS induced by a gate voltage of 5 V is around 0.3-0.4%, however, the in-plane strain transferred onto the FGT layer is as large as 0.5 %. Generally, a large transfer of in-plane strain along out-of-plane direction is very challenging, the author should be careful regarding the estimation of the tensile strain in FGT layer.

Reply: We appreciate the reviewer for the insightful comment. We agree with the reviewer that we should be careful regarding the estimation of strain in the FGT layer. In fact, it is difficult to experimentally quantify the magnitude of the in-plane strain in the FGT. While we strongly believe the observed voltage-induced-redshift of the FGT Raman peaks is indeed a result of lattice expansion, i.e., tensile strain (inferred from FGT Raman peak redshifts with increasing temperature), there is no reference reports that we can use to quantitatively estimate the magnitude of the in-plane strain in the FGT from the amount of the Raman peak redshift. In our manuscript, we might have given the incorrect impression that we experimentally determined a 0.5% tensile strain in our FGT layer. Rather, the 0.5% strain is the quantity we used for the density functional theory (DFT) calculations. The calculated

strain-dependent magnetic anisotropy energy (MAE) of FGT, evaluated for $-0.5 \sim +1.5\%$ strain, shows a more or less monotonic decrease with increasing strain. Our DFT results imply that tensile strain, in general, can act to decrease the MAE of FGT. In the “Theoretical calculations” section (page 14) of the revised manuscript, we have added more data points in the region between 0% and 0.5% to avoid the misunderstanding that we claim a +0.5% strain in the FGT (see revised Figure 4a in main text page 16), and state the general tendency of strain-dependent MAE decrease for tensile strain ($0 \sim +1.5\%$) in the main text. Additionally, in the Raman analysis section of the revised manuscript (page 12), we explicitly state that the exact amount of strain in the FGT cannot be experimentally determined.

We also agree with the reviewer that it is not so trivial to transfer in-plane strain in the out-of-plane direction, particularly between vdW materials. In this regard, we believe the thin polymer (PDMS) layer at the FGT/IS interface in our device might act to mediate the transfer of in-plane strain from the IS to FGT, as already discussed in page 5 of the manuscript. Earlier studies [Nat. Commun. 12, 2018 (2021), Nano Lett. 21, 3956 (2021), Adv. Mater. Inter. 2201463 (2022)] indeed show that polymer layers (e.g., PDMS) can be effectively used to exert in-plane strain in an adjacent vdW/2D material.

List of Changes (highlighted yellow in manuscript):

- Revised discussion in the main text (page 5):

“Nevertheless, this layer acts as a flexible adhesive layer; earlier studies show that polymer layers can effectively transfer in-plane strain to an adjacent vdW material³⁷⁻³⁹, such that it may in fact help transfer in-plane strain from the IS to FGT in this work.”

- Included discussion in the main text (page 12):

“While the FGT Raman peak redshift suggests the presence of tensile strain in FGT, the exact amount of strain in the FGT cannot be estimated due to lack of earlier studies on Raman measurements of the strain state of FGT.”

- Revised discussion in the main text (page 14):

“We find that there is almost no strain dependence in magnetization (Fig. 4a) and also find that the MAE monotonically decreases with increasing strain (Fig. 4a), showing a significant reduction in the MAE even for small strain states ($0 \sim +0.4\%$). These suggest that the reduced FGT magnetic anisotropy, rather than any modulation of its magnetization, is the origin of the observed tensile-strain-induced reduction in H_c .”

Reply to Reviewer #2

Remarks to the Author:

In this manuscript, the authors experimentally demonstrate magnetoelectric interactions in a Van der Waals (vdW)-a material-based device consisting of a ferromagnetic ($Fe_{3-x}GeTe_2$, $x \approx 0.36$) and a ferroelectric (α - In_2Se_3) vdW material. The significant decrease in the coercive field for both positive and negative bias voltages is attributed to the tensile strain-induced reduction of the perpendicular magnetocrystalline anisotropy (MCA), supported by density functional theory calculations. The strain change is from the voltage-induced piezoelectric response on In_2Se_3 . The results show an interesting and important topic of low-power voltage-controlled vdW spintronic devices. Nevertheless, several experiment data may conflict with their conclusion. Thereby, the author should answer the following comments before further consideration.

Reply: We deeply thank the reviewer for the careful reading and critical comments, and particularly appreciate that he/she considers our work to be an interesting and important topic in the growing field of vdW/2D spintronics. Following the constructive comments of the reviewer, we have carried out additional experiments, performed more careful analysis, and included additional discussion to better demonstrate that our experimental data supports our conclusion. In the following, we provide a point-by-point response to the comments and questions, along with the list of necessary revisions made to the manuscript.

1. From Figs 2a, 2b, and S3, the reversible manner of FGT/IS can be observed by applying and withdrawing the voltages. V_G -induced H_c modulation seems reversible and mostly non-remanent within one cycle. The reversibility should be investigated at least 20 cycles to show its stable and reversible manner.

Reply: We thank the reviewer for the important comment. Following the suggestion, we test the reversibility of our measurement over 50 cycles in a FGT/IS heterostructure device, with each cycle consisting of a voltage application ($V_G = 4 \sim 5V$) and returning back to zero bias. We observe that the V_G -induced H_c modulation remains after 50 cycles (see Fig. R4 below), showing that the reversible voltage-tunable magnetic effect is quite robust and stable.

Fig. R4 (SI Fig. S5). The non-remanent and reversible V_G -induced H_c change is consistently observed after 50 cycles of voltage application. The cycle measurements are performed on the FGT/IS device shown in Fig. S2a-c. Each cycle consists of voltage applications up to $V_G = \pm 4-5$ V, and back to $V_G = 0$ V.

List of Changes (highlighted yellow in manuscript):

- Included discussion in the main text (page 7):

“The reversible behaviour is observed over 50 cycles of voltage application (SI Fig. S5), showing that this voltage-tunable magnetic effect is quite robust and stable.”

- Included Fig. R4 (see above) as Fig. S5 and caption in SI (page 7):

2. I suggest the author replenish H_c curves by applying the -4.5 V and -5 V in Fig 2b during negative V_G -induced H_c variation. It is helpful for the reader to understand better.

Reply: We agree with the reviewer that it would be better to include the -4.5 V and -5 V data in Fig 2b. Unfortunately, we do not have those data for the voltage application cycle in the original Fig. 2b. Alternatively, we replace Fig. 2b with a voltage cycle that includes $V_G = -5$ V data (see Fig. R5 below).

Fig. R5 (Fig. 2b). The V_G -dependent M - H loops of the FGT/IS heterostructure for negative V_G .

List of Changes (highlighted yellow in manuscript):

- Revised Fig. 2 and caption in main text (page 8).

Fig. 2. Voltage-dependent magnetic properties measured by MOKE. a-d) The magnetic hysteresis loops of the FGT/IS heterostructure as a function of voltage, for a) positive V_G at 67 K, b) negative V_G at 67 K, c) positive V_G at 80 K, and d) negative V_G at 80 K. The loops are measured as the magnitude of V_G is increased. e) The coercive field plotted as a function of V_G . f) The magnetic hysteresis loops of FGT/h-BN as a function of V_G .

3. The authors declare that the IS surface charge effect can be ruled out, including the current or spin torque effect. Thus, the author needs to give a mechanism explanation about H_c variation issues. For example, from Figs S3, the H_c becomes narrow with approximately the same tendency and shape in FGT under -5 V and 5 V at 80 K, respectively. However, this phenomenon can not be obtained in FGT under -5 V and 5 V at 67 K. What happened? On the other hand, the saturation moments are altered after the reversible test. Why? The magnetic moments should be back to the original position during the manipulation of the ferromagnetism of FGT. Is there another factor left to exclude?

Reply: We appreciate the reviewer for examining our data attentively. As the reviewer points out, the observed voltage effect has some temperature-dependent variations: V_G -dependent H_c change shows some asymmetry at 67 K, while it is more symmetric at 80 K. In addition, the voltage effect shows some minor sample-to-sample variations (see below Fig. R6), with some devices showing more symmetric voltages effects and others showing some non-symmetry. Despite the variations, the overall homopolar V_G -dependent H_c decrease is consistently observed in all the FGT/IS devices investigated.

Fig. R6 (SI Fig. S2). V_G -dependent H_c modulation in other FGT/IS devices.

a) Optical microscopy (OM) image of the FGT(12 nm)/IS(100 nm) device. b) V_G -dependent M - H loops and c) H_c plotted as a function of V_G for the device shown in (a). Measurements at 70 K.

d) OM image of the FGT(10 nm)/IS(40 nm) device. e) V_G -dependent M - H loops and f) H_c plotted as a function of V_G for the device shown in (d). Measurements at 74 K.

g) OM image of the FGT(20 nm)/IS(50 nm) device. h) V_G -dependent M - H loops and i) H_c plotted as a function of V_G for the device shown in (g). Measurements at 80 K.

We suggest the properties of the polymer layer existing at the FGT/IS interface as the possible origin of these temperature-dependent and sample-to-sample variations in the detailed features of the overall voltage-induced H_c decrease. As discussed in the manuscript (page 5) and in the following

replies to comments 4~6, the interfacial polymer at the FGT/IS interface acts as a flexible adhesive layer to transfer in-plane strain between the two vdW materials. Yet, its plastic properties might lead to small variations in the amount of strain transfer for each voltage application, such that there could be minor differences in the voltage dependence of the M - H loops for each measurement cycle. Investigation on the exact role of this interfacial polymer layer is an important topic that requires systematic follow-up studies. Nevertheless, we once again emphasize that a consistent voltage effect, i.e., the homopolar voltage-induced H_c decrease, is always observed in all four FGT/IS heterostructure devices studied (see SI page 4 Fig. S2), such that the main point of our conclusion still holds true.

As for the variation in the MOKE signals upon voltage application, it is understandable to suspect that the magnetization might have been altered during the measurements. However, from our experience measuring vdW FM material flakes using a micro-MOKE set-up inside a low-temperature cryostat, this MOKE signal variation more likely results from technical issues related to the small yet inevitable drift/movement of the measuring location. That is, we might in fact be measuring slightly different regions within the FGT/IS device that might lead to small differences in the MOKE signals. Considering the small lateral size of the FGT/IS heterostructure (5~10 μ m), it is possible that some part of the Gaussian-shaped laser beam can drift outside the heterostructure, i.e., where there is no FGT layer, in the presence of small random sample drifts, leading to reduced MOKE signals (Note that the measuring laser beam spot size (FWHM) is \sim 2 μ m, but due to the Gaussian shape of the beam, signals can be detected up to \sim 5 μ m away from the center). Maintaining the exact same position of the laser beam spot during the measurements is not so trivial with a low-temperature cryostat. While we certainly understand the concerns of the reviewer, we do hope the reviewer recognizes the technical difficulty.

Finally, we rule out any possible current-induced Joule heating effects. In Fig. R8, we plot the saturation Kerr rotation signal with and without bias voltage in two independent FGT/IS devices. We observe that the FGT flakes do not show any significant applied-voltage-dependent change in T_c . If there were any heating effects when voltage is applied, the $V_G = \pm 4/5$ V plots in Fig. R8 would shift left along the T -axis relative to the $V_G = 0$ V plots, which is not the case. This is direct proof that there are no significant current-induced Joule heating effects.

In the revised manuscript, we include possible origins of the MOKE signal variation issues, along with the temperature-dependent variation issues. We thank the reviewer for bringing up these important points that should be discussed in order for the readers to fully comprehend our experimental data.

Fig. R7 (SI Fig. S7). V_G -dependent M - T plots for FGT/IS heterostructure devices. The saturation MOKE signal (Kerr rotation angle) is plotted as a function of temperatures for $V_G = 0$ V and $V_G = \pm 4$ or 5 V. a) M - T plot at $V_G = 0$ V and $V_G = \pm 4$ V in the device shown in Fig. S2a. b) M - T plot at $V_G = 0$ V and $V_G = \pm 5$ V in the device shown in Fig. S2d. The Curie temperature (T_C) of the FGT flakes can be estimated from the temperature at which the MOKE signal (sensitivity $\sim 0.01^\circ$) vanishes. The T_C of the FGT flakes are ≈ 120 K at $V_G = 0$ V, and the T_C shows little change with voltage application ($V_G = \pm 4$ or 5 V) for both FGT/IS devices.

List of Changes (highlighted yellow in manuscript):

- Included discussion in the main text (pages 10-11):

“We note that the overall homopolar voltage effect has some temperature-dependent and sample-to-sample variations (Fig. 2a-e, SI Fig. S2). At 67 K, the voltage-dependent H_c modulation shows some polarity-dependent asymmetry, in contrast to the more or less symmetric voltage-dependent H_c modulation at 80 K. Additionally, the overall voltage-dependent H_c decrease shows some minor sample-to-sample variations (see SI Fig. S2). We suggest the properties of the polymer layer existing at the FGT/IS interface as the possible origin of these temperature-dependent and sample-to-sample variations in the detailed features of the voltage-induced H_c modulation. As discussed earlier, we believe the interfacial polymer at the FGT/IS interface acts as a flexible adhesive layer to transfer strain between the two vdW materials. Yet, there could be small variations in the amount of strain transfer for each voltage-induced-strain-application owing to its plastic properties, resulting in minor differences in the voltage dependence. The exact role of the thin interfacial polymer layer is a topic for follow-up studies.”

- Included discussion in the Methods section of main text (page 20):

“The measuring laser beam spot size (FWHM) is ~ 2 μm on the sample, but due to the Gaussian shape of the beam, signals can be detected > 5 μm away from the center. Considering the small

lateral size of the FGT/IS heterostructure (5 ~ 10 μm), the obtained Kerr rotation is the magnetic signal averaged over a majority of the heterostructure device. Inevitable random sample drifts during the measurements can result in some variation in the Kerr rotation signals.”

- Included discussion in SI (page 6):

“The small change in the MOKE signal (Kerr rotation) before and after voltage application is likely due to sample drifts during the measurements. The MOKE laser beam size is comparable to the heterostructure device size (see Methods in main text).”

- Included Fig. R6 (see above) as Fig. S2 and caption in SI (page 4).

- Included Fig. R7 (see above) as Fig. S7 and caption in SI (page 9).

4. Van der Waals (vdW) materials are strongly bonded two-dimensional (2D) layers bound in the third dimension through weaker dispersion forces. Compared with thin-film-based composite systems, dispersion forces are much smaller than chemical bonds. How to effectively couple the electric and magnetic properties at the ferromagnetic/ferroelectric vdW interface?

Reply: We completely agree with the reviewer that it is not so trivial to couple physical properties between two different vdW materials due to the weak vdW interlayer coupling at the interface. Specifically in our FGT/IS devices, we believe the consistently observed thin polymer layers at the FGT/IS interfaces might act to mediate the transfer of strain from the IS to FGT, as discussed in page 5 of the manuscript. In fact, earlier studies [ref 37~39; Yan et al. *Nat. Commun.* 12, 2018 (2021), Cho et al. *Nano Lett.* 21, 3956 (2021), Song et al. *Adv. Mater. Inter.* 2201463 (2022)] indeed show that polymer layers (PDMS) can be effectively used to exert in-plane strain in an adjacent vdW/2D material.

Likewise, in our FGT/IS devices, in-plane piezoelectric strain in an underlying IS layer can be effectively transferred to the FGT via polymer-mediated interlayer coupling, and hence, the voltage-controlled-piezoelectricity of the IS layer is coupled to the ferromagnetic properties of the FGT layer.

List of Changes (highlighted yellow in manuscript):

- Included discussion in the main text (page 5):

“Nevertheless, this layer acts as a flexible adhesive layer; earlier studies show that polymer layers can effectively transfer in-plane strain to an adjacent vdW material³⁷⁻³⁹, such that it may in fact help

transfer in-plane strain from the IS to FGT in this work.”

5. *The authors claim that “Second, the effect of surface charge induced by the out-of-plane polarization of IS can be excluded since H_c decreases for both positive and negative voltages.” However, the non-symmetric voltage-induced H_c decrease in Figure 2 (c) is inconsistent with the symmetric voltage-induced tensile strain in Figure 3 (b). The polarization-induced surface charge modulation should be considered.*

Reply: We thank the reviewer for carefully examining our data and suggesting a possible mechanism. Considering the ferroelectric nature of IS, we agree that the effect of IS polarization-induced surface charge modulation should be considered to properly interpret our experimental results. The IS polarization, which should switch with voltage polarity, in addition to the IS piezoelectric strain, which should increase with voltage regardless of voltage polarity, can lead to the overall homopolar yet non-symmetric voltage-induced effects. While we include this discussion in the revised manuscript as a possible mechanism (page 9), we believe that the effect of surface charge modulation can be mostly excluded due to the following experimental evidence.

- (1) The ferroelectric surface charge effect would be remanent when the voltage is decreased back to zero. However, in our experiments, the voltage-induced H_c decrease is reversible and mostly non-remanent.
- (2) As discussed in the previous reply to comment 4 and the following reply to comment 6, we observe a thin interfacial polymer layer between the FGT and IS layers in our FGT/IS heterostructure device. This layer likely act to screen surface charges, preventing the effect of IS polarization-induced surface charge modulation.

As in our reply to the previous two comments (3 & 4), we suggest that the non-symmetry observed in the overall homopolar voltage-induced H_c modulation owes to the properties of the polymer layer in between the FGT and IS. That is, the effect of polymer-mediated strain transfer at the microscopic level might not always be identical for each voltage-induced-strain application, such that there could be variations in the voltage dependence, which leads to minor non-symmetry in the overall homopolar voltage effect. This is corroborated by the fact that the voltage effect has some sample-to-sample and temperature-dependent variations, as discussed in our reply to comment 3.

Nevertheless, we once again emphasize that the homopolar voltage-induced H_c decrease is consistently observed in all our FGT/IS heterostructure devices studied, which is the main point of our

work. In the revisions, we include further analysis on the detailed features of the observed voltage-induced H_c modulation effect, along with the possible origin of variations in the voltage dependence.

List of Changes (highlighted yellow in manuscript):

- Included discussion in the main text (page10):

“Given the previous report of remanent effect from the polarization-induced IS surface charge³³, the non-remanent V_G -induced H_c change (SI Fig. S4) provides further evidence that the IS surface charge is not likely responsible for the voltage effect. Furthermore, the polymer layer at the FGT/IS interface would likely act as a charge dissipation layer, screening the IS surface charge from directly affecting the FGT.”

6.Also, deliberately adding a charge dissipation layer underneath the FGT would help to rule out polarization-induced surface charge modulation.

Reply: We thank the reviewer for this suggestion. We completely agree that adding a interfacial layer that may screen the electronic charge could help rule out the effect of IS surface polarization. We argue that the thin polymer layer at the FGT/IS interface (discussed in our replies to comment 3~5) will likely effectively screen surface charges, preventing the IS polarization-induced surface charge modulation to directly affect the FGT. While this polymer layer does not make an ideal vdW interface with the FGT or IS, we find that this layer is consistently observed following our device fabrication process. Thus, this method might provide a practical approach to transfer strain, while simultaneously preventing the effect of surface charge, in vdW heterostructure devices.

List of Changes (highlighted yellow in manuscript):

- Included discussion in the main text (page 10):

“Furthermore, the polymer layer at the FGT/IS interface would likely act to screen the IS surface charge from directly affecting the FGT.”

7.From MAE to H_c , more debates should be added. In addition, the decreased saturation magnetization also will lower the H_c .

Reply: We appreciate the reviewer for this insightful comment. We are fully aware that many material parameters (e.g., MAE, magnetization, magnetic domains, defects, etc.) can affect the H_c . Our explanation that the voltage-dependent H_c change is attributed to the change in MAE might have been too brief, and following the reviewer’s suggestion, we include other possibilities.

We argue that some factors, such as defects, can be quite easily excluded; it is not likely that there will be voltage-induced changes in the defect states of the metallic FGT. We also suggest that any voltage-dependent modulation of the magnetic domains would likely be a consequence of changes in the magnetic anisotropy [Ando et al., *Appl. Phys. Lett.* 109, 022401 (2016)]. Thus, the two dominating factors which could modulate H_c that need to be considered are the magnetic anisotropy and the magnetization [ref. 56, Cullity & Graham, *Introduction to Magnetic Materials*, Chapter 9].

In order to figure out the origin of the voltage/strain-dependent H_c decrease, we carried out additional DFT calculations. Specifically, we evaluated the strain-dependent change in the Fe magnetic moments, in addition to the change in MAE. DFT calculations show that there is almost no strain-dependent change in the magnetization in the FGT, in stark contrast to the large strain-dependent MAE modulation (see Fig. R8 below). This confirms that the strain-induced MAE decrease, rather than reduction in saturation magnetization, as the determining factor for the observed voltage-dependent H_c change. This is in line with our analysis of band structures that the main effect of strain is momentum-dependent shifts in the band energies near the Fermi energy while maintaining the overall exchange splitting.

In the revised manuscript, we include discussion on other material parameters that might cause the H_c decrease. We also add DFT calculation results to show that the strain-dependent modulation of the MAE, rather than the magnetization, as the main origin of our experimental observation.

Fig. R8 (Fig. 4a). MAE/Fe (circles) and magnetization/Fe (triangles) as a function of in-plane strain.

List of Changes (highlighted yellow in manuscript):

- Included discussion in the main text (page14):

“This suggests that the IS piezoelectric strain and accompanied change in lattice constants modulates the H_c of the adjacent FGT. In general, H_c of magnetic materials are determined by many material parameters, including the magnetic anisotropy, magnetization, magnetic domains, defects, etc. However, it is not likely that there will be voltage-induced changes in the defect states of the metallic FGT. We also argue that any voltage-dependent modulation of the magnetic domains would likely be a consequence of changes in the magnetic anisotropy⁵⁵. Thus, it is reasonable to expect the observed reversible voltage-dependent change of H_c can most likely be attributed to a voltage-controlled modulation of either the magnetization or the magnetic anisotropy⁵⁶.”

- Included discussion in the main text (page 14):

“The mechanism of the strain-induced change in the magnetic properties of FGT (e.g., magnetization, magnetic anisotropy) is investigated by first-principles density functional theory”

- Included discussion in the main text (pages 14-15):

“We find that there is almost no strain dependence in magnetization (Fig. 4a) and also find that the MAE monotonically decreases with increasing strain (Fig. 4a), showing a significant reduction in the MAE even for small strain states (0 ~ +0.4%). These suggest that the reduced FGT magnetic anisotropy, rather than any modulation of its magnetization, is the origin of the observed tensile-strain-induced reduction in H_c .”

- Include Fig. R8 (see above) and caption as Fig. 4a in main text (page 16)

8. For some details, the author may condense the abstract. Also, the Scale bar should mark its magnitude in Figures 1 and S1; Add some recent articles in the Introduction.

Reply: We thank the reviewer for the careful reading and constructive suggestions.

List of Changes (highlighted yellow in manuscript):

- We have condensed the abstract from 195 to 147 words (main text page 2).

- The lengths of the scale bars are included inside the figures as well as the figure captions.

- Recent articles (Ref. 9, 10, 11, 12, 20) are cited in the Introduction (main text pages 3-4).

Reply to Reviewer #3

Remarks to the Author:

Voltage control of magnetism is important for developing low-power spintronic devices. Much progress have been done in ferromagnetic/ferroelectric multiferroic heterostructures owing to strain-mediated magnetoelectric coupling. Moreover, the control of magnetism in ferromagnetic/ferroelectric vdW heterostructures has been predicted in many previous papers. This paper is an experimental attempt to realize this aim. Although the change of coercive field by applying voltages is observed, however, the evidence supporting the conclusion is insufficient.

Reply: We deeply appreciate the reviewer for the careful reading and critical comments. As the reviewer points out, this work is the first experimental realization of the theoretically predicted magnetoelectric effect in ferromagnetic/ferroelectric vdW heterostructures. We believe our observation of the voltage control of magnetism provides a significant advance towards the realization of low-power voltage-controlled vdW/2D spintronic devices.

We understand the reviewer's concern that the evidence supporting the conclusion was insufficient in the original manuscript. Following the suggestions of the reviewer, we have conducted all the additional necessary experiments and carried out more in-depth analysis to address the issues. First, we measured the voltage-dependent T_C of the FGT flake in the FGT/IS heterostructure device, and observed that there is no voltage-dependent change in T_C , which rules out the possibility of Joule heating. Second, we show that the voltage effects are not observed in FGT/ β -In₂Se₃ devices, suggesting that the presence of piezoelectric/ferroelectric α -In₂Se₃, is essential for the emergence of the voltage effects. Third, we repeat the experiments in three other independent FGT/IS devices, and find that the homopolar voltage effect is consistently observed. Fourth, we measure that the IS ferroelectric domains are very large, in which case the ferroelectric coercivity will not show a dramatic temperature-dependent change as long as the temperature range is far away from the ferroelectric Curie temperature. Furthermore, we add more analysis and discussion on why we can exclude current-induced spin-torque effects, or any direct interfacial coupling (e.g., FE surface polarization) between the FGT and IS.

We believe the additional data and analysis further support our conclusion of piezo-strain-mediated magnetoelectric coupling in FGT/IS. We once again thank the reviewer for the insightful comments and constructive suggestions which allow us to significantly improve the quality of our work. Below, we provide a point-by-point response to the comments and questions, along with the list of necessary revisions made to the manuscript.

1. The coercive field is $\sim 25\text{mT}$ at 67 K, but the typical H_c for FGT ($\sim 10\text{nm}$) is 2500 Oe (250mT). [Tan, Cheng, et al. *Nature communications* 9, 1554 (2018).

Reply: We understand the concern of the reviewer. The discrepancy arises from the fact that the FGT used in our study is Fe-deficient (hole-doped) $\text{Fe}_{3-x}\text{GeTe}_2$ ($x \approx 0.36$). Compared to stoichiometric Fe_3GeTe_2 , which has large magnetic coercivity (H_c), the hole-doped $\text{Fe}_{3-x}\text{GeTe}_2$ used here has much smaller H_c and lower T_C ; we denote this hole-doped $\text{Fe}_{3-x}\text{GeTe}_2$ as “FGT” in our manuscript. The hole doping (Fe composition) dependent change in the magnetic properties was reported in previous studies [Refs. 25-28; *PRB* 96, 144429 (2017); *PRB* 97, 165415 (2018); *JAP* 120, 083903 (2016); *Nano Lett.* 20, 95 (2020)]. In fact, in our earlier study, it was shown that thin ($\sim 10\text{nm}$) hole doped $\text{Fe}_{3-x}\text{GeTe}_2$ flakes have small $H_c < 30\text{mT}$ [*Nano Lett.* 20, 95 (2020)], much in line with the H_c in this work.

While we already pointed out the use of hole-doped $\text{Fe}_{3-x}\text{GeTe}_2$ in the introduction of the original manuscript (main text page 4), we further emphasize this in the results section in order to highlight that the $\text{Fe}_{3-x}\text{GeTe}_2$ used in our work has much smaller H_c and anisotropy compared to stoichiometric Fe_3GeTe_2 in the revised manuscript.

List of Changes (highlighted yellow in manuscript):

- Included discussion in the main text (page 7):

“Note the small coercive field (e.g., $\approx 30\text{ mT}$ at 67 K) of the FGT ($\text{Fe}_{2.64}\text{GeTe}_2$) used in this work, which is a hole-doped version of Fe_3GeTe_2 .”

2. The mechanism of decrease in coercive field is questionable. More control experiments should be performed. For example, use non-ferroelectric $\beta\text{-In}_2\text{Se}_3$ to replace the ferroelectric $\alpha\text{-In}_2\text{Se}_3$ and perform the same measurements, given that $\alpha\text{-In}_2\text{Se}_3$ and $\beta\text{-In}_2\text{Se}_3$ have similar semiconducting property. The author performed a contrast experiment on FGT/h-BN, but this is not a valid contrast because h-BN is a wide-bandgap insulator, not comparable with $\alpha\text{-In}_2\text{Se}_3$ in terms of electric properties.

Reply: We thank the reviewer for suggestions on additional control experiments that would support our conclusion of *strain-mediated magnetoelectric coupling* in FGT/ $\alpha\text{-In}_2\text{Se}_3$. Following the suggestion, we have carried out voltage-dependent MOKE measurements on FGT/ $\beta\text{-In}_2\text{Se}_3$ heterostructure devices. In Fig. R9 (included in revised SI as Fig. S6), we show that the voltage effects are not observed in FGT/ $\beta\text{-In}_2\text{Se}_3$ devices, suggesting that the presence of $\alpha\text{-In}_2\text{Se}_3$, and in particular

its intercorrelated in-plane and out-of-plane piezoelectricity/ferroelectricity, is essential for the emergence of the voltage effects. Note that the β - In_2Se_3 we used is thicker than the typical α - In_2Se_3 we used for our experiments, and hence, we applied larger voltages up to $V_G = \pm 9$ V.

Fig. R9 (Fig. S6). Voltage-dependent magnetic properties of a FGT(14 nm)/ β - In_2Se_3 (300 nm) heterostructure device. a) OM image of the heterostructure device. b-c) The magnetic hysteresis loops of the FGT/ β - In_2Se_3 heterostructure as a function of voltage, for b) positive V_G , and c) negative V_G . Measurements done at 70 K.

In the following replies to comments 3&4, we also exclude the possibilities of Joule heating, and current-induced spin-transfer-torques and spin-orbit-torques. The former is ruled out by experimental data showing no voltage-dependent T_C change in the FGT, while the latter is excluded by the small current density flowing in the FGT layers which are ~ 5 orders smaller than current densities required for spin-torque effects. See the following replies to comments 3 and 4 for detailed explanation. Furthermore, the presence of an interfacial polymer layer prevents the surface polarization of the ferroelectric α - In_2Se_3 from directly affecting the FGT layer. Thus, the non-remanent and homopolar voltage effect is most likely due to the piezo-strain-mediated magnetoelectric coupling. This is confirmed by the voltage-dependent Raman peak shifts and DFT calculation results.

Finally, we also show that the voltage-induced H_c decrease is reproduced in three other independent FGT/ α - In_2Se_3 heterostructure devices. As seen in the revised SI Fig. S2 (SI page 4; also see Fig. R13 below in reply to comment 10), we find that the voltage-induced H_c decrease is consistently observed in all the α - In_2Se_3 devices studied in this work.

List of Changes (highlighted yellow in manuscript):

- Included discussion in the main text (page 9):

“In another control experiment, MOKE measurements on a FGT/ β - In_2Se_3 heterostructure device,

in which a non-ferroelectric β -In₂Se₃ is used, show no significant V_G -dependent change (SI Fig. S6). These control experiments confirm that the presence of ferroelectric/piezoelectric IS adjacent to FGT is essential for the emergence of the voltage effect on the FGT magnetic properties.”

- Included Fig. R9 (see above) as Fig. S6 and caption in SI (page 8).

- Included Fig. S2 and caption in SI (page 4).

3. The biggest problem is that the author make comparison between data acquired from nonequilibrium state and equilibrium state. Such comparison is not reliable. In the nonequilibrium state where V_g and I_g is not zero, many side effects, not only current-induced effects, but also other spin-orbit coupling effects, can occur, which make the spin system different from that in equilibrium state. I would suggest to insert a thin hBN layer between graphite and In₂Se₃. This would effectively eliminate I_g but maintain the exertion of voltage.

Reply: We thank the reviewer for the critical comment. We agree that when I_G is directly flowing through the FGT, current-induced effects (other than Joule heating) can occur in the nonequilibrium state, the most obvious of which are the spin-transfer-torque and spin-orbit-torque in the metallic FGT. In the following, we present evidence that this is possibly not the case.

- (1) The current density flowing through the FGT/IS junction is $\sim 10^6$ A/m² ($I_G = 40$ μ A at $V_G = -5$ V, and lateral size of FGT/IS junction $\sim 5 \times 10$ μ m²). This is ~ 5 orders of magnitude smaller than the $\sim 10^{11}$ A/m² current density typically required for any significant spin-transfer-torque or spin-orbit-torque effects to occur.
- (2) The interfacial polymer layer situated between the FGT and IS prevents any direct interfacial spin-orbit coupling effects between the two materials.

These observations suggest that current-induced spin-transfer-torque and spin-orbit-torque effects are most likely not the origin of the electrically-tunable magnetic properties. While the equilibrium state control experiment suggested by the reviewer (graphite / FGT / IS / hBN / graphite), would further confirm our point, we do hope the reviewer understands that fabricating multi-layer vdW heterostructure devices (particularly overlapping the layers for sufficient voltage effects) is not so trivial. Given the time required for such experiments, we prefer to perform those in follow-up studies. Nevertheless, we believe our experimental data and above arguments sufficiently demonstrates that current-induced effects can be mostly ruled out (see reply to comment 4 for exclusion of current-induced Joule heating).

List of Changes (highlighted yellow in manuscript):

- Included discussion in the main text (page 9):

“Second, current-induced spin-torque effects can be excluded: the current density flowing through the FGT/IS hetero-junction is $\sim 10^6$ A/m², which is ~ 5 orders of magnitude smaller than the $\sim 10^{11}$ A/m² current density typically required for any significant spin-transfer-torque or spin-orbit-torque effects to occur. Additionally, we do not observe any electrically-induced H_c change in the region of the same FGT/IS heterostructure device where the ferroelectric IS is absent, even when the same level of current is flowing through the region (see SI Fig. S8). This effectively eliminates any possible current-induced spin-torque effect.”

4. The author state that current-induced effects are eliminated, but based on doubtful evidence, which is not convincing enough: (1) I_g is 4 times larger for negative bias than that of positive bias, but H_c seems independent from the voltage polarity. This is a very weak support for the claim. As mentioned by the author in the beginning part, there are some polymer residues on the interfaces, therefore, the current may not distribute evenly on the interfaces. The MOKE measures the magnetic properties only on a small point, not the whole junction. The current density on the measured point may not be 4 times larger as the author announced. (2) The author states that there is no electrically induced H_c change in FGT without ferroelectric. This is not a convincing reason. FGT forms barrier-free Ohmic contact with metal electrodes, whereas in FGT/ In_2Se_3 heterostructure, as shown by Figure 1c, Schottky barrier exists, resulting in possible unevenly distributed heating, making the heating more prominent and not negligible.

Reply: As pointed out by the reviewer, eliminating current-induced effects are required to validate our claim of voltage-induced-piezo-strain effect. Here, we provide additional experimental data and further in-depth discussion in order to exclude any current-induced effects. As discussed in our previous reply to comments 2&3, there are two distinct types of current-induced effects that we should consider: (1) Joule heating, and (2) current-induced spin-torques (e.g., spin-transfer-torque, spin-orbit-torque).

We first directly examine the Joule heating effect by measuring the voltage-dependent T_C of the FGT flake in a FGT/IS heterostructure device. In Fig. R10, we plot the saturation Kerr rotation signal with and without bias voltage in two independent FGT/IS devices. We observe the T_C to be approximately ≈ 120 K for both FGT flakes, and importantly, we do not observe any significant applied-voltage-dependent change in T_C . If there were any heating effects when voltage is applied, the

$V_G = \pm 4/5$ V plots in Fig. R10 would shift left along the T -axis relative to the $V_G = 0$ V plots, which is not the case. This is direct proof that there are no significant current-induced Joule heating effects.

We also note that the MOKE laser beam spot size (FWHM) is ~ 2 μm on the sample, but due to the Gaussian shape of the beam, signals can be detected > 5 μm away from the center. Considering the small lateral size of the FGT/IS heterostructure ($5 \sim 10$ μm), the obtained Kerr rotation is the magnetic signal averaged over a majority of the heterostructure device. Since the MOKE signal is obtained over a large portion of the heterostructure device, it is unlikely that we happen to be measuring at a microscopic position where there is less (or more) electric current flow.

As discussed in our reply to comment 3, we rule out current-induced spin-torque effects due to the low current density flowing in the FGT. This is further supported by the lack of any electrically-induced H_c change in the region of the same FGT/IS heterostructure device where the ferroelectric IS is absent. In SI Fig. S8, we compare the voltage effects in two regions within the same FGT flake. In the FGT/IS heterostructure, V_G -dependent hysteresis loops of FGT that is not on top of IS show little V_G effect (SI Fig. S8a), while V_G -dependent hysteresis loops measured on FGT that overlaps the IS show significant V_G effect (SI Fig. S8b), even though the same level of electric current is flowing through the identical metallic FGT flake. In the original manuscript, the results in Fig. S8 might have been misunderstood to rule out Joule heating effects; in the revised manuscript, we explicitly state that these experiments were in fact performed to exclude spin-torque effects, mentioned in our previous reply to comment 3.

Fig. R10 (SI Fig. S7). V_G -dependent M - T plots for FGT/IS heterostructure devices. The saturation MOKE signal (Kerr rotation angle) is plotted as a function of temperatures for $V_G = 0$ V and $V_G = \pm 4$ or 5 V. a) M - T plot at $V_G = 0$ V and $V_G = \pm 4$ V in the device shown in Fig. S2a. b) M - T plot at $V_G = 0$ V and $V_G = \pm 5$ V in the device shown in Fig. S2d. The Curie temperature (T_C) of the FGT flakes can be estimated from the temperature at which the MOKE signal (sensitivity $\sim 0.01^\circ$) vanishes. The T_C of the FGT flakes are ≈ 120 K at $V_G = 0$ V, and the T_C shows little change with voltage application ($V_G = \pm 4$ or 5 V) for both FGT/IS devices.

List of Changes (highlighted yellow in manuscript):

- Included discussion in the main text (page 9):

“First, current-induced Joule heating effects can be ruled out from the lack of any voltage-dependent change in the T_C of the FGT in the heterostructure devices. In SI Fig. S7, we observe that the T_C of the FGT flakes are ≈ 120 K at zero bias, and the T_C shows little change with voltage application ($V_G = \pm 4 \sim 5$ V).”

- Included Fig. R10 (see above) as Fig. S7 and caption in SI (page 9).

- Included discussion in the main text (page 9):

“Second, current-induced spin-torque effects can be excluded: the current density flowing through the FGT/IS hetero-junction is $\sim 10^6$ A/m², which is ~ 5 orders of magnitude smaller than the $\sim 10^{11}$ A/m² current density typically required for any significant spin-transfer-torque or spin-orbit-torque effects to occur. Additionally, we do not observe any electrically-induced H_c change in the region of the same FGT/IS heterostructure device where the ferroelectric IS is absent, even when the same level of current is flowing through the region (see SI Fig. S7). This effectively eliminates any possible current-induced spin-torque effect.”

- Included discussion in the Methods section of main text (page 20):

“The measuring laser beam spot size (FWHM) is ~ 2 μm on the sample, but due to the Gaussian shape of the beam, signals can be detected > 5 μm away from the center. Considering the small lateral size of the FGT/IS heterostructure ($5 \sim 10$ μm), the obtained Kerr rotation is the magnetic signal averaged over a majority of the heterostructure device.”

5. More importantly, considering the ferroelectric polarization of In_2Se_3 is permanent, the voltage control effect is supposed to be nonvolatile. The non-remanent phenomena the author reported is conflict with this intuitive reasoning. [Huang, Xiaokun, et al. Physical Review B 100.23 (2019): 235445.]

Reply: We appreciate the reviewer’s insightful comment. It has been a tantalizing challenge to manipulate device operations relying on interfacial charge modulation as a result of ferroelectric polarization switching. In our work, the ~ 10 -nm-thick FGT ferromagnetic metallic layer is sufficiently thicker than the Thomas-Fermi screening length, and moreover the interfacial region contains an

unintended ultrathin polymer layer. On these grounds, it is hardly expected that the observed effect is mainly attributed to the interfacial charge modulation due to ferroelectric switching. As pointed by the reviewer, the voltage-induced magnetization change seems to be non-remanent, i.e., volatile. Indeed, we agree with the reviewer's opinion that the observed volatile nature is not compatible to the direct effect of ferroelectric polarization switching. However, we note that the ferroelectric In_2Se_3 material is also piezoelectric, and in particular, has intercorrelated in-plane and out-of-plane piezoelectricity. Considering that the in-plane PFM signal was clearly detected in the material [Cui et al., Nano Lett. 18, 1253 (2018); also see Fig. R11], we know that the effective piezoelectric coefficient d_{31} is significant. It is therefore natural to expect strain-mediated magnetoelectric switching in this device.

Even in this case, we agree with the reviewer that there should be some degree of remanence in the voltage effect due to the remanent piezo-strain still existing at voltages below the ferroelectric coercivity. Since the ferroelectric coercivity of $\alpha\text{-In}_2\text{Se}_3$ is relatively small ≈ 20 mV / nm [Wan et al., Nanoscale 10, 14885 (2018)], we expect a ferroelectric coercivity $\approx \pm 1$ V for the 50-nm-thick In_2Se_3 (see reply to the following comment 6 for more discussion of the ferroelectric coercivity of In_2Se_3 at low temperature). That is, the piezoelectric effect would keep increasing above the ferroelectric coercivity (1 ~ 5 V), but a remanent piezo-strain should still exist within 1 V.

However, in Fig. 2a-e and Fig. 3b, we find that the voltage effects at $V_G = \pm 1$ V are very small, such that it is difficult to clearly discern, within the measurement resolutions, if there are any remanent effects due to ferroelectric polarization. Only when large voltages above the ferroelectric coercivity (e.g., 3 ~ 5 V) are applied, do the voltage-dependent- H_c -change and the Raman-peak-redshift become prominent, suggesting the voltage-induced piezo-strain, rather than ferroelectric polarization, as the origin of the *mostly* non-remanent voltage effect.

We thank the reviewer for allowing us to pay more attention to the useful reference of interfacial coupling induced critical thickness for the ferroelectric bistability of In_2Se_3 . The paper is now cited in the revised manuscript with more discussion on the valuable point. According to the theoretical paper, the critical thickness is three monolayers of In_2Se_3 . Since the thickness of the In_2Se_3 layers we used is 50 ~ 100 nm, our device is free from the critical thickness issue.

List of Changes (highlighted yellow in manuscript):

- Included new reference (Ref. 20; Huang et al. *Phys. Rev. B* 100, 235445 (2019)) in the Introduction section of the main text (page 4).

- Included discussion in the main text (pages 9-10):

“Third, the effect of surface charge induced by the out-of-plane polarization of IS can be excluded since H_c decreases for both positive and negative voltages (Fig. 2a-c). The effect of IS surface charge would be opposite when the voltage polarity is switched. Given the previous report of remanent effect from the polarization-induced IS surface charge³³, the non-remanent V_G -induced H_c change (SI Fig. S4) provides further evidence that the IS surface charge is not likely responsible for the voltage effect. Furthermore, the polymer layer at the FGT/IS interface would likely act to screen the IS surface charge from directly affecting the FGT.”

6. In Line 201, it is mentioned that ferroelectric switching voltage is <1 V for 50-nm-thick In_2Se_3 . However, the author should be aware that this is only applicable for the sample at room temperature. For low temperature (67 K in this work), the situation can be very different, due to the reduced domain kinetics, suppressed thermal assisted switching, et al.

Reply: This is a valid point, and we thank the reviewer for critically pointing this out. The ferroelectric coercive field depends on temperature. In order to experimentally determine the ferroelectric switching voltage of In_2Se_3 at low temperatures, we attempted P - E loop measurements. Unfortunately, the leaky nature of the semiconducting In_2Se_3 prevented us from obtaining the In_2Se_3 low temperature P - E loops. To the best of our knowledge, there are no previous reported studies on the low temperature ferroelectric properties of In_2Se_3 .

In order to estimate the ferroelectric coercivity (E_c) at low temperatures, we refer to the relation between E_c and the saturation polarization P_s which is based on the Landau model

$$E_c = \frac{1}{2} \frac{P_s}{\epsilon \epsilon_0}$$

This relation is based on assumptions that the ferroelectric domains are single domain and that there are no extrinsic effects (e.g. pinning). This approximately holds true for our case, since the In_2Se_3 flakes are exfoliated from single crystals, and also In_2Se_3 exhibit large in-plane domains [see Fig. R11, also refer to Nano Lett. 18, 1253 (2018)]. In this case, since the polarization P_s does not show significant temperature-dependent change unless near the ferroelectric T_C , the ferroelectric coercivity E_c will not show a substantial temperature-dependent change far away from T_C . In fact, this is observed in the temperature-dependent E_c change of other well-known room temperature ferroelectric materials, BiFeO_3 and PZT (see Fig. R12 below). In the work by M. Botea et al. [Electron. Mater. 3, 173 (2022)],

BiFeO₃ shows ~30% increase in coercivity at 80K compared to 300K (Fig. R12a). In the work by Lakeshore Company, PZT(20/80) shows only a two-fold increase in coercivity from 310K to 10 K (Fig. R12b). These studies suggest that despite the obvious increase of coercivity with decreasing temperature, the temperature-dependent change might not be as dramatic. Since the T_C of In₂Se₃ is ~700 K [Phys. Rev. Lett. 120, 227601 (2022)] which is far away from the 70~300 K temperature range, it is highly possible that the applied voltages of $\pm 5 \sim 6$ V, which we used for the MOKE and Raman measurements, are still quite larger than the ferroelectric switching voltage of In₂Se₃ even at ~ 70 K.

Fig. R11. Out-of-plane (OOP; top) and in-plane (IP; bottom) piezo force microscopy (PFM) images of a 50-nm-thick IS flake that we measured. The OOP and IP polarizations are intercorrelated with large domains of lateral size $> 5 \mu\text{m}$, which is comparable to the lateral size of the IS flakes. Measurements at 300 K.

Fig. R12. Temperature-dependent P - V characteristics of (a) BiFeO₃, and (b) PZT(20/80).

Additionally, we once again emphasize that the IS1 and IS2 Raman peak redshifts are homopolar and do not show any remanence, within the measurement resolution, for applied voltages up to ± 6 V. Any large temperature-dependent change in the In_2Se_3 ferroelectric coercivity would manifest as voltage-dependent hysteric behaviour of the Raman peak shifts. The volatile feature leads us to adopt the scenario of strain-mediated electromechanical coupling.

List of Changes (highlighted yellow in manuscript):

- Included discussion in the main text (page 12):

“Note that any remanant in-plane strain at zero bias during the voltage-sweep, due to the remanant ferroelectric polarization, should be minimal considering that the voltage-sweep range (± 6 V) is much larger than the expected ferroelectric coercivity (E_c) of the IS flake (± 1 V)³⁰. While the increase of E_c is expected at low temperatures, other well-known room temperature ferroelectric materials (e.g., BiFeO_3) show 30 - 100 % increase of E_c at cryogenic temperatures compared to room temperature⁵⁴, such that the voltages we applied for the MOKE and Raman measurements are likely larger than the E_c of the IS flake even at 70 K.”

7. In line 165, “67K” should be “67 K”.

Reply: We thank the reviewer for carefully examining our paper. We have corrected this error.

List of Changes (highlighted yellow in manuscript):

- The typo was corrected to “67 K” in main text page 8.

8. What is the T_c of FGT? Does the applied voltage have any effect on the T_c ?

Reply: We thank the reviewer for this critical comment. As seen in our reply to comment 4 and Fig. R10 above, T_c of the FGT flakes in similarly fabricated FGT/IS heterostructure devices were measured to be approximately ≈ 120 K, which is slightly lower than the $T_c \approx 150$ K of the bulk hole-doped FGT. Additionally, we do not observe any significant applied-voltage-dependent change in T_c (Fig. R10). Note that the MOKE signal is averaged over a large portion of the FGT/IS heterostructure device (see reply to comment 4), such that it is unlikely that we happen to be measuring at a microscopic position where there is less/more electric current.

List of Changes (highlighted yellow in manuscript):

- Included discussion in the main text (page 9):

“In SI Fig. S7, we observe that the T_C of the FGT flakes are ≈ 120 K at zero bias, and the T_C shows little change with voltage application ($V_G = \pm 4 \sim 5$ V).”

- Included Fig. R10 (see above reply to comment 4) as Fig. S7 and caption in SI (page 9).

9. From Fig. 1c, the current is about $-40 \mu\text{A}$ at -5 V, indicating a power consumption of $200 \mu\text{W}$. When measuring the MH loops, the voltage is always applied so that a significant Joule heating effect can not be avoided, which will increase the temperature. Please give the comment on effect of Joule effect.

Reply: We agree with the reviewer that we should consider the effect of current-induced Joule heating in the FGT/IS heterostructure device. Indeed, $200 \mu\text{W}$ is a considerable amount of power that can lead to heating effects, particularly at the FGT/IS interface.

However, we believe any current-induced heat generated at the FGT/IS interface is not significant in the metallic FGT layer for the following reason. Any current heating effect would prominently show up as a voltage-dependent decrease of the T_C of the FGT layer, yet, we do not observe any significant voltage-dependent change in T_C , as discussed in our replies to comments 4 & 8 (see Fig. R10 above). It is quite possible that the metallic graphite/FGT top electrode effectively dissipates the current-induced Joule heat, such that it has little effect on the FGT magnetic properties.

List of Changes (highlighted yellow in manuscript):

- Included Fig. R10 (see above reply to comment 4) as Fig. S7 and caption in SI (page 9).

- Included discussion in the main text (page 9):

“First, current-induced Joule heating effects can be ruled out from the lack of any voltage-dependent change in the T_C of the FGT in the heterostructure devices. In SI Fig. S7, we observe that the T_C of the FGT flakes are ≈ 120 K at zero bias, and the T_C shows little change with voltage application ($V_G = \pm 4 \sim 5$ V).”

10. Voltage dependent H_c is asymmetric at 67 K (Fig. 2c) and symmetric at 80 K (Fig. S2). Can author explain the origin of this difference?

Reply: We appreciate the reviewer for examining our data attentively. As the reviewer points out, the observed voltage effect has some temperature-dependent variations: V_G -dependent H_c change shows some asymmetry at 67 K, while it is more symmetric at 80 K. In addition, the voltage effect shows some minor sample-to-sample variations (see below Fig. R13), with some devices showing more symmetric voltages effects and others showing some non-symmetry. Despite the variations, the overall homopolar V_G -dependent H_c decrease is consistently observed in all the FGT/IS devices investigated.

Fig. R13 (SI Fig. S2). V_G -dependent H_c modulation in other FGT/IS devices.

a) Optical microscopy (OM) image of the FGT(12 nm)/IS(100 nm) device. b) V_G -dependent M - H loops and c) H_c plotted as a function of V_G for the device shown in (a). Measurements at 70 K.

d) OM image of the FGT(10 nm)/IS(40 nm) device. e) V_G -dependent M - H loops and f) H_c plotted as a function of V_G for the device shown in (d). Measurements at 74 K.

g) OM image of the FGT(20 nm)/IS(50 nm) device. h) V_G -dependent M - H loops and i) H_c plotted as a function of V_G for the device shown in (g). Measurements at 80 K.

We suggest the properties of the polymer layer existing at the FGT/IS interface as the possible origin of these temperature-dependent and sample-to-sample variations in the detailed features of the

voltage-induced H_c modulation. As discussed in the manuscript (page 5), the interfacial polymer at the FGT/IS interface acts as a flexible adhesive layer to transfer strain between the two vdW materials. Yet, there could be small variations in the amount of strain transfer for each voltage-induced-strain-application owing to its plastic properties, resulting in minor differences in the voltage dependence. Nevertheless, we emphasize that a consistent voltage effect, i.e., the homopolar voltage-induced H_c decrease, despite some variations, is consistently observed in all the FGT/IS heterostructure devices studied, such that the main point of our conclusion still holds true.

In the revised manuscript, we include possible origins of the temperature-dependent variations in the detailed features of the voltage-induced H_c modulation, that are observed at 67 K. We thank the reviewer for bringing up this important point that should be discussed to fully comprehend our experimental data.

List of Changes (highlighted yellow in manuscript):

- Included discussion in the main text (pages 10-11):

“We note that the overall homopolar voltage effect has some temperature-dependent and sample-to-sample variations (Fig. 2a-e, SI Fig. S2). At 67 K, the voltage-dependent H_c modulation shows some polarity-dependent asymmetry, in contrast to the more or less symmetric voltage-dependent H_c modulation at 80 K. Additionally, the overall voltage-dependent H_c decrease shows some minor sample-to-sample variations (see SI Fig. S2). We suggest the properties of the polymer layer existing at the FGT/IS interface as the possible origin of these temperature-dependent and sample-to-sample variations in the detailed features of the voltage-induced H_c modulation. As discussed earlier, we believe the interfacial polymer at the FGT/IS interface acts as a flexible adhesive layer to transfer strain between the two vdW materials. Yet, there could be small variations in the amount of strain transfer for each voltage-induced-strain-application owing to its plastic properties, resulting in minor differences in the voltage dependence. The exact role of the thin interfacial polymer layer is a topic for follow-up studies. Here, we take advantage of the polymer-mediated-strain-transfer to implement voltage controlled magnetoelectric effect in a vdW heterostructure. Furthermore, we emphasize that a consistent voltage effect, i.e., the homopolar voltage-induced H_c decrease, is always observed in all the FGT/IS heterostructure devices studied.”

- Included Fig. R13 as Fig. S2 and caption in SI (page 4).

11. For strain-mediated magnetoelectric coupling in ferromagnetic/ferroelectric multiferroic heterostructures, the magnetism is modulated by piezostain through inverse magnetostriction effect. What is the magnetostriction coefficient of FGT?

Reply: We thank the reviewer for this critical comment. As pointed out by the reviewer, the existence of the inverse magnetostriction effect is a prerequisite for the emergence of magnetoelectric coupling in ferromagnetic/ferroelectric heterostructures.

The magnetostriction coefficient of Fe_3GeTe_2 has not been reported. However, in an earlier study by Yu Wang et al. [*Adv. Mater.* 32, 2004533 (2020)], the presence of the inverse magnetostriction effect in Fe_3GeTe_2 was revealed by probing the strain-dependent modulation of the magnetic properties of Fe_3GeTe_2 directly transferred on a polyimide substrate. A large (150%) increase in the H_c of Fe_3GeTe_2 was reported when an in-plane tensile strain of 0.32% was applied to the polyimide substrate. This study confirms the existence of the inverse magnetostriction effect in Fe_3GeTe_2 .

We note that the tensile-strain-induced H_c decrease that we observe in our work seems opposite to the tensile-strain-induced H_c increase in the report by Yu Wang et al. However, both of these seemingly contradictory tendencies can in fact be well understood by our DFT calculation results. In SI Fig. S10, we see that the MAE decreases with tensile strain (brown plot) for the 1h-doped $\text{Fe}_{2.64}\text{GeTe}_2$ we use in our work (denoted FGT in the manuscript), but the MAE *increases* with strain for lower-level-hole-doping cases (e.g., blue plot for 0.5h) in agreement with the tensile-strain-induced H_c increase in undoped Fe_3GeTe_2 observed by Yu Wang et al.

List of Changes (highlighted yellow in manuscript):

- Included discussion in the main text (pages 17-18):

“We finally note that in an earlier study⁵⁷, the presence of the inverse magnetostriction effect in Fe_3GeTe_2 was revealed by probing the strain-dependent modulation of the magnetic properties of Fe_3GeTe_2 directly transferred on a polyimide substrate. A large increase in the H_c of Fe_3GeTe_2 was reported when an in-plane tensile strain was applied to the polyimide substrate, confirming the existence of the inverse magnetostriction effect in Fe_3GeTe_2 . The tensile-strain-induced H_c decrease that we observe in our work seems opposite to the tensile-strain-induced H_c increase in the study. However, both of these seemingly contradictory tendencies can in fact be well understood by our DFT calculation results. In SI Fig. S10, the MAE decreases with tensile strain for the 1h-doped FGT ($\text{Fe}_{2.64}\text{GeTe}_2$) we use in our work, but the MAE increases with strain for lower-level-hole-

doping cases in agreement with the tensile-strain-induced H_c increase in the un-doped Fe_3GeTe_2 observed by Yu Wang et al.⁵⁷”

12. In Fig. 3a and Fig. S5a, it is hard to find the Raman peak. You'd better plot the Raman peaks at various voltages with a curved line rather than a straight line. How do you get the error bar in Fig. 3b and Fig. S5b? what is the resolution of Raman spectroscopy?

Reply: We thank the reviewer for examining our data attentively. In order to better monitor the voltage-dependent change in the Raman peak positions, in the revised Fig. 3a (see Fig. R14 below), we plot the IS1, IS2, FGT1, and FGT2 Raman peak positions at different V_G 's as gray dotted lines and the Raman peak positions at zero bias as black solid lines, such that the voltage-dependent Raman peak shifts can be better visualized. We believe this is a better way to show the peak position change compared to a curved line.

We note that the FGT Raman peak positions are obscured by the numerous Raman peaks which overlap in the measuring range (80 to 160 cm^{-1}). In order to quantify the Raman peak positions, the Raman spectra was fitted using the multiple-Voigt-function model; the peaks of the fitted IS and FGT Raman peaks are marked with the gray dotted lines below.

The error bars in Fig. 3b and Fig. 5b were obtained in the following way. For each measurement, the Si substrate Raman peak position should ideally be fixed. Yet, in reality, there are inevitable fluctuations in the Si peak positions. In our experiments, we indeed observe small fluctuations in the Si Raman peak positions; this could be due to simple measurement errors, or small changes in the calibration. This implies that there will be inevitable fluctuations in the IS or FGT Raman peak positions, which we define as the error in determining the IS or FGT Raman peak positions. Moreover, the error in determining the peak position should also be proportional to the full-width-at-half-maximum (FWHM) of each peak. Thus, using the measured Si peak fluctuation, Si peak FWHM, and FGT or IS peak FWHM, we can estimate

$$\text{Error bar} = (\text{FGT or IS peak fluctuation}) = (\text{Si peak fluctuation}) \times \frac{(\text{FGT or IS peak FWHM})}{(\text{Si peak FWHM})}$$

The spectral resolution of the micro-Raman (beam spot $\approx 2 \mu\text{m}$) spectroscopy we used is 1 cm^{-1} . The Raman peak positions are determined by fitting the measured data using the multiple-Voigt-function-model. While the Raman peak shift at low voltages are small, at $V_G = \pm 6\text{V}$, the FGT and IS Raman peaks show shifts around 1~2 cm^{-1} .

Fig. R14 (Fig. 3a). Voltage-dependent Raman shifts. a) A series of Raman spectra of the FGT/IS heterostructure as a function of V_G , measured at 70 K. The four fitted Raman peak positions at zero bias are shown as solid black lines and the peak positions for each voltage application is shown as gray dotted lines.

List of Changes (highlighted yellow in manuscript):

- Revised Fig. 3a (see Fig. R14 above) and caption in main text (page 13):
- Included discussion in the Methods section of main text (pages 20-21):

“The spectral resolution is 1 cm^{-1} . The Raman peak positions are determined by fitting the measured data using the multiple-Voigt-function-model. In our experiments, we observe small fluctuations in the Si Raman peak positions; this could be due to simple measurement errors, or small changes in the calibration. This implies that there will be inevitable fluctuations in the IS or FGT Raman peak positions, which we define as the error in determining the IS or FGT Raman peak positions. Thus, using the measured Si peak fluctuation, Si peak FWHM, and FGT or IS peak FWHM, we can estimate

$$\text{Error bar} = (\text{FGT or IS peak fluctuation}) = (\text{Si peak fluctuation}) \times \frac{(\text{FGT or IS peak FWHM})}{(\text{Si peak FWHM})} \quad \text{”}$$

REVIEWER COMMENTS

Reviewer #1 (Remarks to the Author):

The authors have properly clarified all my concerns, thus I support the publication of this paper in Nature Communications.

Reviewer #2 (Remarks to the Author):

The authors have addressed all the comments raised by the reviewers. The revised manuscript now can be accepted in Nature Communications. Besides, the manuscript may be further improved based on the suggestion below:

- (1) An ultrathin conductive metal layer may be a better dissipation layer than PDMS polymer layer.
- (2) Variables such as the subscript “x” in $\text{Fe}_{3-x}\text{GeTe}$ should be used in italics.
- (3) The statement “the magnetization per Fe” is not rigorous enough. Since the magnetization itself represents the magnetic moment per unit volume/atom/mass.

Reviewer #3 (Remarks to the Author):

I appreciate the author's efforts in conducting control experiments to address our concerns. The experiments show that the voltage-controlled effect disappears when In_2Se_3 is replaced with non-ferroelectric materials, which substantially confirms the role of ferroelectricity in the observed phenomenon. It is obvious that the MH loops at 0 V in Fig. S6b and S6c are different. Moreover, the MH loops at ± 9 V are also different. However, the author states that “In another control experiment, MOKE measurements on an FGT/ β - In_2Se_3 heterostructure device, in which a non-ferroelectric β - In_2Se_3 is used, show no significant VG-dependent change (SI Fig. S6).” This statement is not right.

While some of my concerns have been addressed, the origin of the observed phenomenon remains uncertain, primarily due to commonly recognized limitations of strain transfer across vdW gaps. Van der Waals force is the weakest chemical force, so it appears highly challenging to induce the transfer of in-plane tensile strain from the 2D ferroelectric to the FGT across the van der Waals gap. Theoretically, we can compare the force required to induce 0.5% strain with the maximum force that vdW interaction can provide. The strength of van der Waals forces across the OOP direction ranges from 0.4-4.0 kJ/mol, while the IP shear forces are several orders lower. By contrast, the Young's modulus of typical 2D materials is ~ 300 GPa for monolayers [npj Comput Mater 4, 49 (2018)], and increases cubically with thickness [Advanced materials 24.6 (2012): 772-775]. With theoretical calculation, we can estimate that the force required to induce strain is several orders of magnitude greater than the force provided by vdW forces. This theoretical discrepancy impedes the conclusions drawn in the paper.

In particular, in the revised manuscript, the thickness dependence was not observed. “The voltage effect does not show FGT-thickness-dependence up to 20 nm”. This poses a significant obstacle in reaching the conclusion. The author acknowledges this in their response and proposes a plausible explanation that residual polymer mediates strain

transfer. This explanation appears reasonable since polymer can indeed form stronger chemical bonds with neighboring materials than vdW forces. However, this hypothesis lacks supporting experimental evidence. The author should not present residual polymer as an inevitable factor to readers. In fact, it is easy to prepare a residue-free interface using dry transfer methods. For example, one can use PDMS to lift hBN flakes and then contact the bottom surface of hBN with FGT via van der Waals forces. The hBN/FGT layer is then placed on top of In₂Se₃, and finally, the hBN layer can be removed using reactive ion etching. A comparison between samples with and without polymer residues can validate the proposed assumption.

In addition, in the revised manuscript, the author states that residual polymer can screen charges. I think this statement is inappropriate. Only grounded metals can screen charges induced by ferroelectric polarization, and the polymer residues are acting as a dielectric material and would actually lead to the accumulation of interface charges.

The history of the applied electric field is important for strain-mediated magnetoelectric coupling. Please clarify the sequence of the applied electric fields.

The detail on how to fit Raman spectra should be given. The reference on multiple-Voigt-function model also needs to be cited. The Raman spectra in Fig. S9a is also needed to be fitted and indicates the Raman peaks.

On page 6 in the response letter, "We also note that this result is insensitive to the fitting functions; the same voltage-dependent redshift is also observed when the Raman spectra are fitted using the pseudo-Voigt functions." Please show the data.

In Fig. S5, please provide some typical MH loops at ± 4 V and 0 V after several cycles. It is better to plot all the H_c at ± 4 V and 0 V to demonstrate the robust and stable modulation.

The scale bar in Fig. S1 should be nm instead of μm .

In Fig. 2a and 2b, the saturation Kerr rotation signal is not same.

In Fig. 2, S2, S4 and S8, when increasing the applied voltages, the saturation Kerr rotation signal decreases if H_c has a change and does not change if H_c does not change. Can the author give an explanation? Does the voltage decrease the saturation magnetization? It seems to be the heat effect.

Response to the reviewer's comments

We sincerely thank the reviewers for their time and effort in thoroughly evaluating our manuscript for the second time. We appreciate that Reviewers 1 and 2 support the publication of the manuscript in *Nature Communications*. In this revision, we include additional experimental data and further analysis & discussion, which addresses the remaining concerns of Reviewer 3. The constructive suggestions by the reviewers have substantially improved the quality of this work, and hence, we believe that the revised manuscript is now suitable for publication in *Nature Communications*. Below we provide a point-by-point response to the reviewers' comments.

Reply to Reviewer #1

Remarks to the Author:

The authors have properly clarified all my concerns, thus I support the publication of this paper in Nature Communications.

Reply: We once again thank the reviewer for the careful reading and critical comments, which make us significantly improve our manuscript.

Reply to Reviewer #2

Remarks to the Author:

The authors have addressed all the comments raised by the reviewers. The revised manuscript now can be accepted in Nature Communications. Besides, the manuscript may be further improved based on the suggestion below:

Reply: We once again sincerely thank the reviewer for the attentive reading and critical comments. The revised manuscript is further improved, following the additional suggestions of the reviewer.

(1) An ultrathin conductive metal layer may be a better dissipation layer than PDMS polymer layer.

Reply: We thank the reviewer for bringing up an important point that a conductive metal layer may be a better charge dissipation layer. Following the comment, we remove the statement that the residual

polymer can screen charges, and mention that a thin metallic layer might be a more effective charge dissipation layer. We also add a brief discussion that the relatively small polymer-induced-interfacial-charge is quite possibly not large enough to have a significant effect on the FGT magnetic properties.

List of Changes (highlighted yellow in manuscript):

- Remove statement on polymer-charge-screening, and add discussion in the main text (page 10):

“We also note that while the interfacial polymer layer might act as a dielectric resulting in accumulated interfacial charges induced by the IS polarization, the relatively thick metallic FGT thickness (~10 nm), compared to the electron screening length, would prevent large electron/hole doping effects despite the possible presence of accumulated interface charges; earlier studies show that extremely large electron/hole doping, rarely achievable in solid state devices, is required for significant changes in the FGT magnetic anisotropy²⁸. Deliberately adding a thin metallic charge dissipation layer underneath the FGT might help further rule out the effect of polarization-induced surface charge modulation.”

(2) *Variables such as the subscript “x” in Fe_{3-x}GeTe should used in italics.*

Reply: We thank the reviewer for carefully examining our manuscript. We have changed the variable *x* into italic fonts in the title, abstract, and main text.

List of Changes (highlighted yellow in manuscript):

- Changed the variable *x* into italic fonts in the title (page 1), abstract (page 2), and text (pages 4 & 14).

(3) *The statement “the magnetization per Fe” is not rigorous enough. Since the magnetization itself represents the magnetic moment per unit volume/atom/mass.*

Reply: We thank the reviewer for this important comment. We have corrected this mistake by changing the term “*magnetization per Fe*” to “*magnetic moment per Fe*” where appropriate.

List of Changes (highlighted yellow in manuscript):

- Changed the term “magnetization per Fe” to “magnetic moment per Fe” in the text (page 14) and figure caption (page 17).

Reply to Reviewer #3

Remarks to the Author:

I appreciate the author's efforts in conducting control experiments to address our concerns. The experiments show that the voltage-controlled effect disappears when In₂Se₃ is replaced with non-ferroelectric materials, which substantially confirms the role of ferroelectricity in the observed phenomenon.

Reply: We highly appreciate the careful examination, comments, and suggested experiments by the reviewer during these two rounds of revisions, which guide us to significantly improve the quality and presentation of our work. We hope that the revised manuscript provides enough additional data, analysis, and discussion to alleviate the concerns of the reviewer.

It is obvious that the MH loops at 0 V in Fig. S6b and S6c are different. Moreover, the MH loops at ± 9 V are also different. However, the author states that "In another control experiment, MOKE measurements on an FGT/ β -In₂Se₃ heterostructure device, in which a non-ferroelectric β -In₂Se₃ is used, show no significant V_G -dependent change (SI Fig. S6)." This statement is not right.

Reply: We thank the reviewer for bringing up this critical point. We admit that we were not very clear on our explanation about the FGT/ β -In₂Se₃ experiments. The FGT/ β -In₂Se₃ heterostructure device shown in Fig. S6a-c consists of a large FGT flake that happens to have many stepped regions, and Fig. S6b and Fig. S6c are, in fact, measured on different regions within the FGT flake. This explains why the M - H loops are already different at 0 V. Unfortunately, we do not have data for both positive and negative V_G on the same FGT region in this particular device.

In order to leave no doubt that FGT/ β -In₂Se₃ does not show any voltage effect, we repeat the measurements on another FGT/ β -In₂Se₃ heterostructure device, the results of which are shown in Fig. R1. We clearly observe that this device shows no significant V_G -dependent change of the M - H loops for both positive/negative V_G . These two control experiments with the non-ferroelectric β -In₂Se₃ indeed confirms the essential role of the ferroelectric/piezoelectric α -In₂Se₃ in the observed magnetoelectric effect.

In the revised manuscript, we include Fig. R1 as SI Fig. S6d and S6e. We also include a brief explanation why the M - H loops in Fig. S6b and c are different at 0 V. Furthermore, we slightly modify and soften our statement on the non-existent V_G -dependent change in FGT/ β -In₂Se₃.

Fig. R1. a) OM image of a FGT(12 nm)/ β -In₂Se₃(60 nm) heterostructure device. b) The V_G -dependent magnetic hysteresis loops of the device. The plot legends are in sequential order.

List of Changes (highlighted yellow in manuscript):

- Include Fig. R1 (see above) as Fig. S6d and S6e and caption in SI (page 8).

- Add discussion in SI Fig. S6 caption SI (page 8):

“The FGT flake in this device has many stepped regions and b) and c) are measured on different regions within the FGT flake.”

- Modify discussion in main text (page 9):

“does not show significant change when V_G is applied”

While some of my concerns have been addressed, the origin of the observed phenomenon remains uncertain, primarily due to commonly recognized limitations of strain transfer across vdW gaps. Van der Waals force is the weakest chemical force, so it appears highly challenging to induce the transfer of in-plane tensile strain from the 2D ferroelectric to the FGT across the van der Waals gap. Theoretically, we can compare the force required to induce 0.5% strain with the maximum force that vdW interaction can provide. The strength of van der Waals forces across the OOP direction ranges from 0.4-4.0 kJ/mol, while the IP shear forces are several orders lower. By contrast, the Young's modulus of typical 2D materials is ~ 300 GPa for monolayers [npj Comput Mater 4, 49 (2018)], and increases cubically with thickness [Advanced materials 24.6 (2012): 772-775]. With theoretical calculation, we can estimate that the force required to induce strain is several orders of magnitude greater than the force provided by vdW forces. This theoretical discrepancy impedes the conclusions

drawn in the paper. In particular, in the revised manuscript, the thickness dependence was not observed. "The voltage effect does not show FGT-thickness-dependence up to 20 nm". This poses a significant obstacle in reaching the conclusion.

The author acknowledges this in their response and proposes a plausible explanation that residual polymer mediates strain transfer. This explanation appears reasonable since polymer can indeed form stronger chemical bonds with neighboring materials than vdW forces. However, this hypothesis lacks supporting experimental evidence. The author should not present residual polymer as an inevitable factor to readers. In fact, it is easy to prepare a residue-free interface using dry transfer methods. For example, one can use PDMS to lift hBN flakes and then contact the bottom surface of hBN with FGT via van der Waals forces. The hBN/FGT layer is then placed on top of In₂Se₃, and finally, the hBN layer can be removed using reactive ion etching. A comparison between samples with and without polymer residues can validate the proposed assumption.

Reply: We thank the reviewer for the critical comment and constructive suggestion. As acknowledged by the reviewer as a plausible explanation, we postulate that the interfacial polymer layer acts to mediate the strain transfer. Indeed, earlier studies (*Adv. Mater.* **32**, 2004533 (2020), *Nat. Commun.* **12**, 2018 (2021), *Nano Lett.* **21**, 3956 (2021), *Adv. Mater. Interfaces* **9**, 2201463 (2022), etc.) show the effectiveness of polymer-mediated-strain-transfer in vdW materials. In particular, it has been reported that H_c changes by 150% with an applied strain of 0.32% in a flexible FGT/polymide device (*Adv. Mater.* **32**, 2004533 (2020)), as discussed in our manuscript (page 18).

Nevertheless, we completely agree with the reviewer that a comparison between samples with and without polymer layers would provide direct proof of this assumption. Following the suggestion, we attempted various dry transfer methods in order to fabricate heterostructures without any interfacial polymer layer. Specifically, we use various types of polymer materials (PC, PPC, gel-pak, PDMS) for the dry transfer (pick-up-and-release) method. In the end, we were unsuccessful in preparing residue-free samples. Typically, releasing the vdW material cleanly from the polymer during the dry transfer method requires a heating process. The difficulty lies in that we need to limit the heating when we handle FGT; the ferromagnetic properties of FGT disappear after high-temperature (>80°C) annealing. We are also reluctant to apply chemical and/or etching methods which we find alters the ferroelectric and ferromagnetic properties of the In₂Se₃ and FGT, respectively. Due to these restrictions, we were not able to prepare polymer-free interfaces with the dry transfer method. We note that this residue issue seems more severe for In₂Se₃ compared to other vdW materials (e.g., h-BN, in which residue-free surfaces have been successfully prepared by the dry transfer method using gel-pak).

Interestingly, we find that the voltage effect on the FGT magnetic properties is consistently observed regardless of whichever polymer material we use for the dry transfer process. Fig. R2 shows TEM images and the V_G -dependent magnetic hysteresis loops of the FGT/ In_2Se_3 heterostructure device, fabricated by dry transfer using gel-pak. We find (1) the existence of a uniformly thin interfacial polymer (Fig. R2a), and (2) the V_G -dependent H_c decrease (Fig. R2b-d), much like the samples shown in our manuscript which are fabricated using PDMS. We note that both PDMS and gel-pak are adhesive flexible materials typically used for dry transfer of vdW materials.

In the revised manuscript, we acknowledge that a comparison between samples with and without interfacial polymer layer is required to fully validate our proposed assumption. We also explain why it is not trivial to fabricate FGT/ In_2Se_3 devices without interfacial polymer layers, and specifically suggest investigating the exact role of the thin interfacial polymer layer as a follow-up study. While we cannot provide control experiment data on heterostructure devices without interfacial polymer layers at this time, we once again emphasize that the consistently observed voltage effect demonstrates that our FGT/ In_2Se_3 device provides an effective and practical platform to realize the magnetoelectric effect in vdW material systems.

Fig. R2. a) The cross-sectional TEM images of FGT/IS heterostructure devices fabricated by dry-transfer using gel-pak. We see the consistent presence of residual polymer layer at the FGT- In_2Se_3 interface. b) Optical microscopy (OM) image of the FGT/IS device fabricated using gel pak. c-d) V_G -dependent $M-H$ loops for c) positive V_G , and d) negative V_G . V_G -dependent H_c decrease is observed. The plot legends are in sequential order.

List of Changes (highlighted yellow in manuscript):

- Add section “Discussion: role of the interfacial polymer layer in the strain-mediated magnetoelectric effect” in main text (pages 17-18).

- Add discussion in main text (page 18):

“Full validation of the proposed strain-mediated mechanism requires further investigation, and we propose follow-up studies. A comparison between samples with and without interfacial polymer layers can reveal the exact role of the thin interfacial polymer layer. However, the removal of this interfacial layer has been difficult (see Methods). Nevertheless, we obtain consistently similar polymer thickness under our sample fabrication conditions (see SI Fig. S11). Here, we take advantage of the polymer-mediated-strain-transfer to implement voltage controlled magnetoelectric effect in a vdW heterostructure, and emphasize that a consistent voltage effect, i.e., the homopolar voltage-induced H_c decrease, is always observed in all the FGT/IS heterostructure devices studied.”

- Add discussion in Methods (pages 20-21):

“In addition to PDMS, we use various types of polymer materials (PC, PPC, gel-pak) for this dry transfer method, and as discussed in the main text, we always find the presence of an interfacial polymer layer. Typically, releasing the vdW material cleanly from the polymer requires a heating process. The difficulty lies in that the heating process should be limited when handling FGT; the ferromagnetic properties of FGT disappear after high-temperature ($>80^\circ\text{C}$) annealing. Additionally, applying chemical or etching methods, which alters the ferroelectric and ferromagnetic properties of the IS and FGT, respectively, should be avoided. Due to these restrictions, we are not able to prepare polymer-free interfaces with the dry transfer method. Nevertheless, the voltage effect on the FGT magnetic properties is consistently observed regardless of the polymer material used for the fabrication process (see SI Fig. S11 for sample fabricated by the dry transfer method using gel-pak).”

- Include Fig. R2 (see above) as Fig. S11 and caption in SI (page 13).

In addition, in the revised manuscript, the author states that residual polymer can screen charges. I think this statement is inappropriate. Only grounded metals can screen charges induced by ferroelectric polarization, and the polymer residues are acting as a dielectric material and would actually lead to the accumulation of interface charges.

Reply: We thank the reviewer for this important point. We agree that the polymer layer cannot screen charges and following the comment, we remove the incorrect statement that the residual polymer can screen charges. Nevertheless, the non-remanent and homopolar nature of the consistently observed voltage effect still supports the fact that the In_2Se_3 ferroelectric polarization is most likely not the reason for the voltage-induced H_c -decrease. Furthermore, we also note that the relatively thick metallic FGT thickness (~ 10 nm), compared to the electron screening length, would prevent large charge/hole doping effects despite the possible presence of accumulated interface charges.

In the revised manuscript, in addition to retracting our statement on polymer charge screening, we also add a brief discussion that the relatively small polymer-induced-interfacial-charge is quite possibly not large enough to have a significant effect on the FGT magnetic properties. We also mention that a thin metallic layer might be a better charge dissipation layer.

List of Changes (highlighted yellow in manuscript):

- Remove statement on polymer-charge-screening, and add discussion in the main text (page 10):

“We also note that while the interfacial polymer layer might act as a dielectric resulting in accumulated interfacial charges induced by the IS polarization, the relatively thick metallic FGT thickness (~ 10 nm), compared to the electron screening length, would prevent large electron/hole doping effects despite the possible presence of accumulated interface charges; earlier studies show that extremely large electron/hole doping, rarely achievable in solid state devices, is required for significant changes in the FGT magnetic anisotropy²⁸. Deliberately adding a thin metallic charge dissipation layer underneath the FGT might help further rule out the effect of polarization-induced surface charge modulation.”

The history of the applied electric field is important for strain-mediated magnetoelectric coupling. Please clarify the sequence of the applied electric fields.

Reply: We thank the reviewer for pointing out the importance of the history of applied electric field in the magnetoelectric effect. Following the suggestion, we clarify the voltage application sequence. All the plot legends in the MOKE data (Figs. 2, S2, S4, S6, S7, S8) are now in sequential order. The MOKE measurements are done as we progressively increase or decrease the magnitude of V_G . For instance, in Fig. 2a, the measurements are done in the following order: $0\text{V} \rightarrow 1\text{V} \rightarrow 2\text{V} \rightarrow \dots \rightarrow 5\text{V}$. Note that all the MOKE data presented are measured as the magnitude of V_G is increasing, with the one exception being

Fig. 2c. Figure 2c shows data that is measured as V_G is returning back to zero bias, i.e., V_G is progressively changed from -5V to 0V. In fact, as seen in Fig. R3 below, the same trend is observed when the V_G -decrease-sequence (Fig. R3a) is compared to the V_G -increase-sequence (Fig. R3b). That is, the voltage effect is reversible.

We also note that the Raman measurements in Fig. 3 are performed as the magnitude of V_G is progressively increased ($0\text{ V} \rightarrow +6\text{ V}$, and $0\text{ V} \rightarrow -6\text{ V}$). After each V_G application, Raman measurements are performed at zero bias, the results of which are shown in SI Fig. S9.

Fig. R3. The magnetic hysteresis loops of the FGT/IS heterostructure as a function of V_G for the device shown in main text Fig. 1 and 2. MOKE measurement data as a) magnitude of V_G is being decreased (data partially duplicated from Fig. 2c), and b) magnitude of V_G is being increased. The plot legends are in sequential order.

List of Changes (highlighted yellow in manuscript):

- Add description on the voltage application sequence in Fig. 2 caption (page 8):

“The plot legends are in sequential order, i.e., the measurements are performed as the magnitude of V_G is progressively increased or decreased.”

- The plot legends in Fig. 2c, 2f, S2b, S2e, and S2h have been modified to be in chronological order (main text page 8 and SI page 4).

- Add description on the voltage application sequence in Fig. 3 caption (page 13):

“The measurements are performed as the magnitude of V_G is progressively increased ($0\text{ V} \rightarrow +6\text{ V}$, and $0\text{ V} \rightarrow -6\text{ V}$). After each V_G application, Raman measurements are repeated at zero bias (see SI Fig. S9).”

- Add description on the voltage application sequence for the MOKE measurements in SI Fig. S2, Fig. S4, Fig. S6, and Fig. S8 captions (SI pages 4, 6, 8, and 10).

“The plot legends are in sequential order.”

The detail on how to fit Raman spectra should be given. The reference on multiple-Voigt-function model also needs to be cited. The Raman spectra in Fig. S9a is also needed to be fitted and indicates the Raman peaks. On page 6 in the response letter, “We also note that this result is insensitive to the fitting functions; the same voltage-dependent redshift is also observed when the Raman spectra are fitted using the pseudo-Voigt functions.” Please show the data.

Reply: We appreciate the reviewer’s curiosity about details on fitting our Raman spectra. The Raman data in Fig. 3 and Fig. S9 are fitted with multiple-peak Voigt functions (using Origin 2020 software). The Voigt function is a convolution of Lorentzian and Gaussian functions, and has been widely used to fit Raman spectra in many earlier studies (*J. Appl. Phys.* **113**, 211301 (2013).; *Nat. Commun.* **8**, 96 (2017).; *Sci. Rep.* **11**, 3361 (2021), etc.). Multiple peaks are simultaneously fitted, and the best-fit parameters (positions, heights, and widths) are found. The peak positions determined from the best-fit are plotted in Fig. 3b and Fig. S9b. We include these details in the revised Methods section.

Following the suggestion of the reviewer, we also mark the Raman peak positions in the Raman spectra of the remanent state, i.e., when the voltage is removed, as gray dotted lines in Fig. S9a (see Fig. R4a below). We hope this better visualizes that the voltage-induced Raman peak redshifts are non-remanent.

Fig. R4. Raman shifts at zero bias after voltage application. a) A series of zero-bias Raman spectra of the FGT/IS heterostructure after applying the V_G in parenthesis, measured at 70 K. The four fitted Raman peak positions at zero bias are shown as solid black lines and the peak positions for each voltage application is shown as gray dotted lines. b) A summary of the zero-bias Raman peak positions shown in a), with the horizontal axis indicating the applied V_G before measuring the Raman shifts at zero-bias.

Finally, fitting results based on the pseudo-Voigt function fitting are shown in Fig. R5. Pseudo-Voigt functions are also widely used to fit Raman spectra in many other studies (*J. Phys. Chem. Lett.* **9**, 4294 (2018).; *Sci. Rep.* **10**, 4516 (2020).; *Sci. Rep.* **13**, 2725 (2023).). The pseudo-Voigt function we use, which is essentially a linear combination of Lorentzian and Gaussian functions, is as follows:

$$y = y_0 + A \left[m_u \frac{2}{\pi} \frac{w}{4(x - x_c)^2 + w^2} + (1 - m_u) \frac{\sqrt{4 \ln 2}}{\sqrt{\pi} w} e^{-\frac{4 \ln 2}{w^2} (x - x_c)^2} \right]$$

The Raman peak positions with voltage application clearly show V_G -dependent redshift (Fig. R5a), while the redshift disappears when the voltage is returned back to zero (Fig. R5b). This is the same trend seen in Fig. 3b and Fig. S9b of our manuscript, which shows that the same V_G -dependent effect is extracted using both the Voigt-function-model and the pseudo-Voigt-function-model.

Fig. R5. Summaries of V_G -dependent Raman shifts from pseudo-Voigt function fit a) when voltage is applied, and b) when the voltage is removed.

List of Changes (highlighted yellow in manuscript):

- Add discussion on details of Raman peak fitting in Methods (page 22):

“The Raman peak positions are determined by fitting the measured data using the multiple-Voigt-function-model^{58,59}. Multiple peaks are simultaneously fitted and the best-fit parameters (positions, heights, and widths) are found. The peak positions determined from the best-fit are plotted in Fig. 3b and Fig. S9b”

- Replace Fig. S9b and caption with Fig. R4b (see above) in SI (page 11).

- Include references 58 and 59 in main text (page 26):

In Fig. S5, please provide some typical MH loops at ± 4 V and 0 V after several cycles. It is better to plot all the H_c at ± 4 V and 0 V to demonstrate the robust and stable modulation.

Reply: We thank the reviewer for suggesting a more persuasive way to present our data. Following the suggestion, we add two representative M - H loops from our V_G -dependent MOKE measurements, and also plot a few more H_c 's at $V_G = \pm 4$ V and 0 V in Fig. R6. Please note that we could not measure V_G -dependent M - H loops for every cycle due to the large amount of time required for 50 cycles; we typically need ~ 1 hour to measure V_G -dependent M - H loops for a cycle, such that it is unreasonable to measure every cycle. We do agree with the reviewer's suggestion to include as many H_c 's as possible, but we hope the reviewer understands our limited circumstances.

Fig. R6. a) The non-remanent and reversible V_G -induced H_c change is consistently observed after 50 cycles of voltage application. The cycle measurements are performed on the FGT/IS device shown in Fig. S2a-c. Each cycle consists of voltage applications up to $V_G = \pm 4$ -5 V, and back to $V_G = 0$ V. b-c) The magnetic hysteresis loops with $V_G = 0$ V and $V_G = -4$ V for b) the 1st cycle and c) the 50th cycle further demonstrate the stability of the device.

List of Changes (highlighted yellow in manuscript):

- Replace Fig. S5 and caption with Fig. R6 (see above) in SI (page 7).

The scale bar in Fig. S1 should be nm instead of μm .

Reply: We thank the reviewer for the careful examination. We have corrected this mistake.

List of Changes (highlighted yellow in manuscript):

- Changed the scale bar units from " μm " to "nm" in Fig. S1a,b (SI page 2).

In Fig. 2a and 2b, the saturation Kerr rotation signal is not same. In Fig. 2, S2, S4 and S8, when increasing the applied voltages, the saturation Kerr rotation signal decreases if H_c has a change and does not changes if H_c does not changes. Can the author give an explanation? Does the voltage decrease the saturation magnetization? It seems to be the heat effect.

Reply: We appreciate the reviewer for examining our data attentively. As the reviewer points out, the observed Kerr rotation angle (θ_K) indeed shows some V_G -dependent variations. It is understandable to suspect that the magnetization might have been altered during V_G application. However, we respectfully argue that the V_G -dependent change in θ_K is a measurement-to-measurement variation issue rather than a V_G -dependent decrease of the saturation magnetization. Please note that we provided data in the previous revision (SI Fig. S7) which mostly rules out the Joule heating effect; the measurement-to-measurement variation is represented by the error bars in SI Fig. S7.

To be specific, while some measurements show V_G -dependent decrease of θ_K (e.g., Fig. 2a,c,d), other measurements do not show V_G -dependent θ_K change (e.g., SI Fig. S4b,c,d. also see Fig. R3b above); note that the legend in Fig. S4 is in chronological order, such that, for instance, the θ_K does not change as V_G changes from 0 V (red) to 5 V (green) in Fig. S4c. In fact, in Fig. S4, we find that the θ_K value at 0 V increases after voltage application in some cases (compare the red and blue curves in Fig. S4a and S4c).

In the following, we explain our understanding of the variation in the MOKE signals upon voltage application. From our experience measuring vdW FM material flakes using a micro-MOKE set-up inside a low-temperature cryostat, the observed θ_K variation likely results from technical issues related to the small yet inevitable drift/movement of the measuring location. That is, we might in fact be measuring slightly different regions within the FGT/IS device that might lead to differences in the θ_K signals. Considering the small lateral size of the FGT/IS heterostructure (5~10 μm), it is possible that some part of the Gaussian-shaped laser beam can drift outside of the heterostructure, i.e., where there is no FGT layer, in the presence of small random sample drifts, leading to reduced θ_K signals. Note that the measuring laser beam spot size (FWHM) is $\sim 2\mu\text{m}$, but due to the Gaussian shape of the beam, signals can be detected up to $\sim 5\mu\text{m}$ away from the center. Maintaining the exact same position of the laser beam spot during the measurements is not so trivial with a low-temperature cryostat. While we certainly understand the concerns of the reviewer, we do hope the reviewer recognizes the technical difficulty. Considering this measurement-to-measurement variation issue, in SI Fig. S7 (V_G -dependent M vs. T plots), we plot the θ_K value averaged over 3~10 measurements.

List of Changes (highlighted yellow in manuscript):

- Add discussion in Methods (page 21):

“Yet, it is possible that some part of the Gaussian-shaped laser beam can drift outside of the heterostructure, i.e., where there is no FGT layer, in the presence of inevitable random sample drifts, which can result in some variation in the Kerr rotation signals. The MOKE data are averaged over 3~10 measurements in order to compensate, to some degree, for the measurement-to-measurement variation issue.”

REVIEWERS' COMMENTS

Reviewer #2 (Remarks to the Author):

The authors have addressed all the comments raised by the reviewer. The revised paper can now be accepted in Nature Communications.

Reviewer #3 (Remarks to the Author):

We are happy to recommend this paper for publication as adequate efforts have been made by the authors.

Response to the reviewer's comments

We sincerely thank the reviewers for their time and effort in thoroughly evaluating our manuscript for the third time. We appreciate that all three Reviewers support the publication of the manuscript in *Nature Communications*.

Reply to Reviewer #2

Remarks to the Author:

The authors have addressed all the comments raised by the reviewer. The revised paper can now be accepted in Nature Communications.

Reply: We once again thank the reviewer for the careful reading and critical comments, which make us substantially improve the quality of this work.

Reply to Reviewer #3

Remarks to the Author:

We are happy to recommend this paper for publication as adequate efforts have been made by the authors.

Reply: We deeply appreciate the reviewer for all the critical comments in the three rounds of reviews. The revised manuscript has been significantly improved following the constructive suggestions of the reviewer.